# High- and low-temperature pyrolysis profiles describe volatile organic compound emissions from western US wildfire fuels

**Kanako Sekimoto[1,2,3,‡], Abigail R. Koss[1,2,4,*,‡], Jessica B. Gilman[1], Vanessa Selimovic[5], Matthew M. Coggon[1,2], Kyle J. Zarzana[1,2], Bin Yuan[1,2,6], Brian M. Lerner[1,2,†], Steven S. Brown[1,4], Carsten Warneke[1,2], Robert J. Yokelson[5], James M. Roberts[1], Joost de Gouw[1,2,4]**

[1] NOAA Earth System Research Laboratory (ESRL), Chemical Sciences Division, Boulder, CO 80305, USA

[2] Cooperative Institute for Research in Environmental Sciences, University of Colorado Boulder, Boulder, CO 80309, USA

[3] Graduate School of Nanobioscience, Yokohama City University, Yokohama, Kanagawa 236-0027, Japan

[4] Department of Chemistry and Biochemistry, University of Colorado Boulder, Boulder, CO 80302, USA

[5] Department of Chemistry, University of Montana, Missoula, MT 59812, USA

[6] Institute for Environment and Climate Research, Jinan University, Guangzhou, China

[*] Now at Department of Civil & Environmental Engineering, Massachusetts Institute of Technology, Cambridge, MA 02142, USA

[†] Now at Aerodyne Research, Inc., Billerica, MA 01821, USA

[‡] K. Sekimoto and A. Koss are equally contributing first authors.

*Correspondence to*: Kanako Sekimoto (sekimoto@yokohama-cu.ac.jp)

**Abstract.** Biomass burning is a large source of volatile organic compounds (VOCs) and many other trace species to the atmosphere, which can act as precursors to secondary pollutants such as ozone and fine particles. Measurements performed with a proton-transfer-reaction time-of-flight mass spectrometer during the FIREX 2016 laboratory intensive were analyzed with Positive Matrix Factorization (PMF), in order to understand the instantaneous variability in VOC

emissions from biomass burning, and to simplify the description of these types of emissions.
Despite the complexity and variability of emissions, we found that a solution including just two
emission profiles, which are mass spectral representations of the relative abundances of emitted
VOCs, explained on average 85% of the VOC emissions across various fuels representative of
the western US (including various coniferous and chaparral fuels). In addition, the profiles were
remarkably similar across almost all of the fuel types tested. For example, the correlation
coefficient $r^2$ of each profile between Ponderosa pine (coniferous tree) and Manzanita (chaparral)
is higher than 0.84. The compositional differences between the two VOC profiles appear to be
related to differences in pyrolysis processes of fuel biopolymers at high and low temperatures.
These pyrolysis processes are thought to be the main source of VOC emissions. "High-
temperature" and "low-temperature" pyrolysis processes do not correspond exactly to the
commonly used "flaming" and "smoldering" categories as described by modified combustion
efficiency (MCE). The average atmospheric properties (e.g. OH reactivity, volatility, etc) of the
high- and low-temperature profiles are significantly different. We also found that the two VOC
profiles can describe previously reported VOC data for laboratory and field burns.


**1 Introduction**
Biomass burning is a large source of volatile organic compounds (VOCs) and other trace
species to the atmosphere. Reactions involving these VOCs produce ozone and fine particles,
which are important air pollutants and radiative forcing agents (Alvarado et al., 2009; Alvarado
et al., 2015; Yokelson et al., 2009; Jaffe et al., 2012). Some VOCs from fires also have direct
health effects (Naeher et al., 2007; Roberts et al. 2011). Biomass burning occurs in wildfires,
controlled burns of wildland and agricultural fuels, and in residential wood stoves and industrial
processes. Given the variety of fuels and burning conditions, it is unsurprising that the VOC
composition of biomass burning emissions varies greatly between different fire states, locations,
and studies. Therefore, it is important to understand VOC emissions from biomass burning in
detail and develop a predictive capability that explains some of the variability in VOC emissions.
Multiple complex processes take place in biomass burning, including (i) distillation with
release of water vapor and terpenes, (ii) pyrolysis of solid biomass giving off flammable gases,
(iii) flaming combustion, and (iv) non-flaming processes loosely lumped with smoldering
combustion such as glowing (gasification) of biomass (Yokelson et al., 1996; Yokelson et al.,
1997; Collard and Blin, 2014; Liu et al., 2016). The main source of VOC emissions is pyrolysis
of the polymers that form biomass such as cellulose, hemicellulose, and lignin. The temperature
of the reaction and the physical characteristics of the biopolymer control which pyrolysis
mechanism (e.g. depolymerization, fragmentation, or aromatization) is the main source of
emitted VOCs (Yokelson et al., 1996; Yokelson et al., 1997; Collard and Blin, 2014; Liu et al.,
2016). In a given fire, the processes (i)-(iv) occur simultaneously, but the relative importance of
each process and temperature can change with time, which relates to the variability in integrated
VOC emissions between different fires. This variability is often parameterized as a function of
modified combustion efficiency (MCE = $\Delta CO_2/(\Delta CO+\Delta CO_2)$) (Yokelson et al., 1996). $CO_2$ and
CO are representative gases emitted from the flaming and smoldering combustion processes,
respectively, and are measured in most biomass burning studies. MCE is generally higher in
flaming combustion (> 0.9) and lower in smoldering combustion (< 0.9) (Akagi et al., 2011).

The National Oceanic and Atmospheric Administration (NOAA) led the Fire Influence on

Regional and Global Environments Experiment (FIREX) 2016 laboratory intensive conducted at
the U.S. Forest Service Fire Sciences Laboratory in Missoula, Montana to study emissions of
trace gases and aerosol from wildfires. Emissions from various fuels representative of the
western U.S. were sampled under controlled conditions by extensive instrumentation
(https://www.esrl.noaa.gov/csd/projects/firex/firelab/instruments.html). Experiments included
so-called stack burns, in which emissions from an evolving burn were entrained into a large-
diameter stack and sampled by various instruments. VOCs were measured by several instruments,
including a PTR-ToF-MS (proton-transfer-reaction time-of-flight mass spectrometer) which
captured gas-phase emissions with a fast time response during stack burns. The measurements
show variability in VOC composition as the fire shifts between a dynamic mix of distillation,
pyrolysis, flaming combustion, and "smoldering" combustion (here we use smoldering as a
rough term to include various "non-flame" processes such as gasification). Ions measured with
the PTR-ToF-MS were interpreted using a combination of gas-chromatographic pre-separation
experiments, literature review, time-series analysis, and comparison to other instruments (Koss et
al., 2018). Approximately 90% of the instrument signal could be attributed to identified VOCs.

The aims of this work are to understand the variation in gas-phase emissions both over the

course of a fire and on a fire-integrated basis. Ultimately, this improved understanding of
emissions variability could be used to simplify predictions of the emission of secondary organic
aerosol (SOA) and ozone precursors. To do this, the VOCs observed by PTR-ToF-MS in stack
burns were analyzed using positive matrix factorization (PMF). We show that much of the
observed variability in VOCs can be explained by only two factors, and that these two factors are
qualitatively related to the temperature of the pyrolysis processes, which are the main sources of
the VOC emissions from biomass burning. Based on this result, the two factors are named as a
high-temperature pyrolysis factor and a low-temperature pyrolysis factor. The two factors are
compared between fuels. Importantly, the high-temperature factor is quantitatively similar
between different fuels, and the same is true for the low-temperature factor. The VOCs present in
each factor are discussed in terms of composition, reactivity with OH, and propensity to form
secondary organic aerosol. The relative importance of high- and low-temperature pyrolysis
factors is quantified for each fuel and discussed with respect to physical properties of the fuel
and the burn dynamics. We also investigate how well VOC emissions in biomass burning can be
modeled by the two PMF emission profiles through comparisons with previously reported data
from laboratory burns and wildfires. Finally, emissions of some specific compounds are
discussed.


**2 Methods**
**2.1 VOC measurements by PTR-ToF-MS**
Fire emissions were measured during the FIREX 2016 intensive at the Fire Sciences
Laboratory in Missoula, Montana. The facility consists of a large combustion chamber and has
been described in detail previously (Christian et al., 2003; Christian et al., 2004; Burling et al.,

2010).

VOC measurements were performed using several instruments, including a PTR-ToF-MS.
This instrument employed a high-resolution ToF mass analyzer (Aerodyne Research Inc, MA,
USA; Tofwerk AG, Thun, Switzerland) and measured with a time resolution of 2 Hz. VOCs and
some inorganic compounds were ionized by proton transfer from $H_3O^+$ reagent ions. We include
the inorganic compounds in the discussion of VOCs. Species with a proton affinity higher than
that of water can be measured, which includes many unsaturated and polar compounds. The mass
resolution of the instrument (3000-5000 FWHM m/Δm) was sufficient to determine the
elemental composition of ions and separate many isobaric compounds. Before each fire,
background air in the combustion chamber was measured directly for several minutes. The
instrument has been described in detail by Yuan et al. (2016; 2017), and operation, calibration,
and peak identification during the FIREX 2016 laboratory intensive were described by Koss et al.

(2018).


**2.2 Fuel and biomass burn descriptions**
Fifteen types of natural fuel mixtures, most of which are representative of important western
U.S. ecosystems, were burned (Table 1). The names below are largely taken from the dominant
plant species: (i) Ponderosa pine, (ii) Lodgepole pine, (iii) Loblolly pine, (iv) Douglas fir, (v)
Engelmann spruce, (vi) Subalpine fir, (vii) Juniper, (viii) Bear grass, (ix) Ceanothus, (x)
Chamise-contaminated, (xi) Chamise-uncontaminated, (xii) Manzanita-contaminated, (xiii)
Manzanita-uncontaminated, (xiv) Sagebrush, and (xv) Excelsior (aspen wood shavings).
"Contaminated" chaparral fuels (Manzanita and Chamise) were collected from a heavily air-
polluted site near San Dimas, CA, while "uncontaminated" fuels were collected from a cleaner
site in North Mountain, CA. Individual components of various fuel complexes, including canopy,
litter, duff, and rotten wood, were also burned separately. Fuel moisture content ranged from
0.6% to 55.6%, and instantaneous MCE ranged from 0.75 to 1. Additional details on the fires
and fuels are given by Selimovic et al. (2018) including: pre- and post-fire weight, weight of fuel
components, and elemental composition (C, H, N, S, and Cl by weight). Each fuel type was
burned several times. All fires consumed most of the fuel. The present experiments did not have
a direct measurement of temperature within the fire, which is not homogeneous and therefore
difficult to define. Rather, the air temperature of the emissions was measured by the FTIR
instrument, located at the sampling inlet of the PTR-ToF-MS. The hot gases from the fire were
mixed with air from the room, cooling the air significantly, but the trends in temperature are
related to the initial temperature of the emitted gases.

**2.3 PMF analysis**
Data from 51 burns measured by PTR-ToF-MS (Table 1) were analyzed using positive
matrix factorization (PMF), a numerical method that can be used to determine major
compositional categories of emissions, their compositional profiles, and their relative
enhancements over time. PMF was conducted using the PMF Evaluation Tool v. 2.08A (Ulbrich
et al., 2009). The basic principles of PMF and application to atmospheric chemistry
measurements have been previously described (Ulbrich et al., 2009; Paatero and Tapper, 1994;
Paatero, 1997).
More than 1000 ions were quantified in the PTR-ToF-MS mass spectra between $m/z$ 12-217.
Of these, 574 were selected for PMF analysis (Table S1). These 574 ions were resolved from
neighboring peaks, were enhanced during at least one fire, and exclude primary (e.g., $H_3O^+$ and
$H_3O^+(H_2O)$) and contaminant ions (e.g., Teflon fragments and transition metals) (Koss et al.,
2018). The ion signals (in units of normalized counts-per-second; ncps), which are normalized to
the $H_3O^+$ ion intensities and corrected for ToF-duty cycle, humidity dependence, and $H_3O^+$ ion
depletion as described by Koss et al. (2018), were analyzed using PMF. Typically, raw ion
signals in units of "counts-per-second (cps)" have been used for PMF analysis. However, cps
VOC ion signals are affected by temporal variability (depletion and instability) in primary ion
intensity and humidity during the fire. To obtain PMF results that exclude instrument effects, the
normalized and corrected ion signals are used in this analysis. The uncertainties of the
normalized and corrected ion signals were calculated based on those originating from the raw
(cps) ion signals. We chose to use instrument signal rather than mixing ratio because many ion
masses cannot be unambiguously related to a single VOC contributor: they have several
contributors, or result from fragmentation, and cannot be converted to mixing ratio. For example,
$C_7H_{13}^+$ ($m/z$ 97.101) is a fragmentary product ion of at least five different VOCs, whose relative
contributions are different between fires. However, variability in these ion signals still contains
information useful for PMF. To interpret the PMF results, we did convert to mixing ratio where
possible (Section 2.4). 528 compounds were quantified, of which 156 are identified VOCs. The
PTR-ToF-MS measures 50-80% of total emitted non-methane VOC mass, with uncertainty in
this value due to semivolatile compounds (Hatch et al., 2017).
In this work, we applied PMF to extended time series, in which all fires of a particular fuel
type (e.g., Ponderosa pine) were consolidated into a single data matrix (Figure S1), as well as
time series of single fire data. Each fuel type was burned several times. Some individual fires of
a particular fuel did not necessarily capture the full possible range of high- and low-temperature
fire conditions, because of variability in the relative amounts of fuel parts, fuel moisture content,
when fuel was added, or other differences. PMF using the consolidated time series makes it
possible to capture the widest possible range of fire conditions. This approach also simplifies the
comparison of average emission profiles between different types of fuels. Details on preparation
of ion signal and uncertainty datasets are described in the Supporting Information (S1).

The discussion in Section 3 is based on the 2-factor PMF solutions. Out of the 574 ions, 434

ions were fitted well and together represented 99% of the total ion signal. A total of 140 ions
were not well fitted as the difference between their measurements and the PMF reconstruction
was higher than 50%; these ions are excluded from the factors presented here. Ulbrich et al.
(2009) suggest that poor retrieval of ions with less than 5% of total signal is not uncommon.

**2.4 Calculations of OH reactivity and volatility**

To characterize key chemical properties of the emission profiles derived from PMF analysis,

we compare the OH reactivity and volatility of VOCs in each profile. These calculations require
conversion of the emission profiles from instrument signal (ncps) to mixing ratio (ppbv).
Fragment ions, cluster ions, and ions not well fitted by PMF were excluded from the 574 ions
used in PMF analysis and calibration factors were applied to the remaining 400 ions to convert
them to mixing ratio. Of these, 156 have known VOC contributors, and account for 90% of the
total instrument signal of non-primary and non-contaminant ions between *m/z* 12-217. (This
corresponds to an average of 92% of the total VOC concentration detected by PTR-ToF-MS).
Details on identification of the VOC contributors to ion masses and calibration are described by
Koss et al. (2018).

We quantified the importance of the 156 identified ions to OH chemistry by multiplying the

VOC + OH reaction rate coefficient ($cm^3$/molecule/sec) with the VOC fraction in the profile
(ppbv VOC/ppbv of total VOC emitted) with a scaling factor to convert from VOC molar
emission (ppbv VOC) to number density (molecule/$cm^3$ at experimental conditions of 900 mbar
and 26°C). The resulting OH reactivity is in units of per second per ppbv of total VOCs
measured with PTR-ToF-MS (1/sec/ppbv of total VOC emitted). For ions with more than one
contributor, a weighted average rate constant was determined. Rate constants were taken from
the literature (Atkinson and Arey, 2003; NIST Chemical Kinetics Database; Cicerone and
Zellner, 1983; Gilman et al., 2015) or estimated from structurally similar VOCs. Details can be
found elsewhere (Koss et al., 2018).
We also quantified volatility using the saturation concentration at 25 ºC ($C_0$, µg m$^{-3}$).
Saturation concentrations were taken from the literature (Rumble, 2017-2018; NIST Chemistry
WebBook; Yaws, 2015) where possible, and otherwise estimated based on the elemental
composition of the ion (Li et al., 2016). Volatility determined from elemental composition is
uncertain, especially for compounds with very low volatility where the uncertainty can be several
orders of magnitude (Li et al., 2016). We determined volatility for the 400 non-fragmentary ions.
We define volatility bins as follows, after Li et al. (2016): volatile organic compounds (VOC, $C_0$
> 3×10$^6$ µg m$^{-3}$), intermediate volatility compounds (IVOC, $300 < C_0 < 3×10^6$ µg m$^{-3}$), and
semivolatile compounds (SVOC, $0.3 < C_0 < 300$ µg m$^{-3}$). Separation into such volatility bins is
commonly used as an aid to discussion of SOA formation potential and gas/particle partitioning
(Donahue et al., 2011).


**3 Results and discussion**
**3.1 Two-factor parameterization of VOC emissions from biomass burning**
Figure 1a shows the time series of selected VOC ion signals from burning a representative
mixture of Ponderosa pine fuels. In these lab fires, total VOC emissions (red line in Figure 1a)
often increase immediately and substantially during the initial combustion (for 170 seconds after
starting the burn in this example), and then total emissions gradually decrease as the flames die
out. Emissions of individual VOCs can be seen to fall into two categories: (i) higher emissions
during the first part of the fire, e.g. naphthalene, which correlates with the PMF factor we will
largely attribute below to high-temperature pyrolysis (blue line in Figure 1a), and (ii) higher
emissions during the latter part of the fire, e.g., syringol, which correlates with the PMF factor
we will attribute below to low-temperature pyrolysis (green line in Figure 1a). This separation
into two categories is typical for most fires, with a few exceptions discussed later (e.g., burns of
duff and rotten wood).
These two PMF factors (Figure 1b) describe the total VOC emissions remarkably well for
most fuels: residuals (the differences between the measured ion signals and the calculated ion
signals based on the PMF fits) are less than 15% on average, except for Douglas fir, Engelmann
spruce, and Subalpine fir for which the residual average is 20-25%. The residuals for individual
fuels are summarized in Table 1c. For most of the fuels, the time series of the first and second

factors are strongly correlated with those of naphthalene and syringol, respectively (correlation coefficient ($r^2$) > 0.74). On the contrary, emissions of compounds mainly from flaming or non-pyrolysis smoldering processes, such as CO, $CO_2$, and $NO_x$ (Figure 1c), do not correlate well with the individual PMF factors (more detailed discussion is given in Section 3.5). This indicates that the two PMF factors do not correspond to the flaming and smoldering combustion processes that are described by MCE and often referenced in biomass burning literature. The main source of VOC emissions is pyrolysis of fuel biopolymers, and *not* the flaming and/or other combustion processes. Therefore, we primarily attribute these two factors to high-temperature pyrolysis and low-temperature pyrolysis, respectively, and will use these names to describe these factors in this work. Our association between the factors and pyrolysis temperature is related more rigorously to the distribution of products observed as a function of pyrolysis temperature in the next section. When allowing more than two factors in PMF, the time series and mass spectral profiles of the additional factors can be represented as an "intermediate" or "splitting" of high- and/or low-temperature factors which can be described by a linear combination of the two factors. As examples, Figures S2 and S3 show the correlation between *n*-factor solutions (*n* = 3, 4) and PMF results from high- and low-temperature factors for Ponderosa pine datasets. This suggests that only two factors, i.e., high- and low-temperature pyrolysis factors, were needed to explain most of the variability we observed for the VOC emissions from biomass burning.

There are notable exceptions to the two-factor solution, including an infrequently observed, but important, third factor that we call a "distillation" factor, and a fourth profile observed during burns of duff. Several fires contain a distillation phase, in which a brief burst of VOCs, typically enriched in terpenes, is emitted immediately prior to ignition. However, PMF captured this phase for only a limited number of burns in which the distillation phase contained sufficient gas-phase emissions and lasted long enough (~30 seconds). When a two-factor solution is used, the terpenes are largely grouped with the high-temperature pyrolysis factor. Duff is defined as a "layer of moderately to highly decomposed leaves, needles, fine twigs, and other organic material found between the mineral soil surface and litter layer of forest soil" (Reardon, 2007). The duff PMF solutions have residuals larger than 80% when solved with only two factors. This means that duff burns have a unique VOC emission pattern that cannot be explained by only high- and low-temperature factors. These exceptions are discussed in more detail later.

**3.2 VOC emission profiles of high- and low-temperature pyrolysis factors**

The mass spectral profiles of the relative abundances of emitted VOCs for the individual PMF factors obtained from a given fuel type are similar for replicate burns of the same fuel type. When comparing the PMF profiles for two individual burns of the Ponderosa pine realistic mixture, the correlation coefficient ($r^2$) is higher than 0.92 for both the high- and low-temperature pyrolysis factors (Figure 2a). Importantly, the mass spectra for the high-temperature pyrolysis factor are also very similar between different fuels, and the same is true for the low-temperature pyrolysis factor. For example, the correlations of each profile between (i) Douglas fir and Ponderosa pine, (ii) Manzanita (chaparral) and Ponderosa, and (iii) Bear grass and Ponderosa have a slope near 1 and $r^2 \geq 0.83$ (Figures 2b-d). In contrast, the correlation between the high- and low-temperature mass spectra is visually clearly lower ($r^2 < 0.69$, Figure 2e). Figure 3 shows the average VOC emission profiles of the two factors obtained using PMF results of 15 different fuels. The fractions of individual ion peaks in the emission profiles are summarized in Table S1. These average profiles are in good agreement with profiles of individual fuels: a best fit of $0.96 <$ slope $< 1.04$ and $r^2 > 0.84$, except for high-temperature factor of Excelsior with $r^2 = 0.68$ (Table 1d and Figure S4). Excelsior is an unusual fuel in that it consists of fine shavings of a single fuel component (wood). VOC composition in high- and low-temperature profiles is discussed in Section 3.3.1.

The compositional differences between the two profiles can be qualitatively explained by the temperature of the pyrolysis reactions thought to be the main production mechanism of the VOCs, such as depolymerization, fragmentation, and aromatization (Yokelson et al., 1996; Yokelson et al., 1997; Collard and Blin, 2014; Liu et al., 2016). This is illustrated by the relative contributions from the high-temperature versus low-temperature factors for most emitted VOCs. VOCs expected from high-temperature processes have a higher emissions contribution from the high-temperature factor, and likewise for low-temperature VOCs and the low-temperature factor.

Figure 4a shows the contribution of each factor to selected pyrolysis products from major fuel biopolymers, i.e., hemicellulose, cellulose, and lignin. The contributions of individual VOCs are expressed by their normalized fractions ($F_{high-T}$ and $F_{low-T}$) of high- and low-temperature factors: $F_{high-T} = Fraction_{high-T}/(Fraction_{high-T} + Fraction_{low-T})$ and $F_{low-T} = Fraction_{low-T}/(Fraction_{high-T} + Fraction_{low-T})$, where $Fraction_{high-T}$ and $Fraction_{low-T}$ correspond to fractions (in ppbv/total VOC ppbv) of individual species in the high- and low-temperature VOC profiles

(Figure 3), respectively. Figure 4b also shows the relationship between pyrolysis temperature and
representative products for individual biopolymers as reported in the literature (Collard and Blin,
2014). During the heating of biomass, different chemical bonds within the biopolymers are
broken, which results in the release of VOCs and in rearrangement reactions within the matrix of
the residue. Low temperature pyrolysis breaks the bonds between the monomer units of the
polymers. Depolymerization in lignin (300-500 °C) produces guaiacols, (iso)eugenol, and
syringol. Furans and furfurals are dominantly formed from cellulose and hemicellulose (300-
400 °C). Emissions of these compounds have a larger contribution from the low-temperature
factor ($F_{\text{low-T}}$ = 60-100%). Higher temperatures allow reaction of functional groups and covalent
bonds in polymers and monomers. The resulting fragmentation emits various VOCs: for example,
hydroxyacetone, acetaldehyde, and acetic acid from depolymerization of cellulose and/or
hemicellulose. These VOCs have roughly equal contributions from low- and high-temperature
factors. The release of oxygenated compounds during depolymerization and fragmentation
increases the carbon percentage of the residual biopolymers. Benzene rings and aromatic
polycyclic structures form, which is termed char. Higher temperature pyrolysis breaks
progressively stronger bonds in char (> 500 °C). This aromatization process gives off aromatic
compounds with short substituents (e.g., phenol), non-substituted aromatics (e.g., benzene), and
polycyclic aromatic hydrocarbons (PAHs such as naphthalene). Most of those aromatics have a
large contribution from the high-temperature factor ($F_{\text{high-T}}$ = 60-100%). As the temperature
increases, substituents of the aromatic rings disappear and PAHs are dominantly produced. This
is consistent with the contribution of the high-temperature factor to phenol ($F_{\text{high-T}}$ = 60%),
benzene (77%), and naphthalene (92%).
These many diverse chemical processes are likely happening simultaneously during a fire,
and their relative intensities may change based on fuel composition, fuel moisture content, or
other as-yet poorly defined parameters. However, the net result of all these variables is the
emission of just two major compositional groups. The VOCs that comprise these two groups
mostly consist of the pyrolysis products described above and their analogs. During most of these
fires, the emissions of any particular VOC can be described by a linear combination of the high-
temperature and low-temperature pyrolysis time series. Some VOCs are emitted mainly from the
high-temperature pyrolysis; some mainly from the low-temperature profile; and others have a
mixed contribution. This is quantified by $F_{\text{high-T}}$ as described above. We sorted the VOCs by
$F_{high-T}$, to show how the chemical composition of emissions changes from high- to low-
temperature pyrolysis process. Figure 5 shows the chemical characteristics of compounds that
are mostly emitted in the high-temperature pyrolysis ($F_{high-T}$ = 80-100% in panel (a)), mostly
emitted in the low-temperature pyrolysis ($F_{high-T}$ = 0-20% in panel (e)), or have mixed
contributions from both pyrolysis ($F_{high-T}$ = 60-80% in panel (b), 40-60% in (c), and 20-40% in
(d)). $F_{high-T}$ of each individual VOC is shown in Figure S5. In the category emitted mostly by the
high-temperature pyrolysis, important compounds include alkyl-substituted aromatics and
aliphatic alkenes (Figures 5a and b), whereas carbonyls have more equal contributions from the
high- and low-temperature pyrolysis processes. It should be noted that terpenes (e.g.,
(oxygenated) monoterpenes and isoprene) emitted from distillation are grouped with the high-
temperature pyrolysis (Figures 5a and b; Section 3.6).

Several nitrogen (N)-containing compounds also fall into high or low temperature categories

consistent with behavior previously reported in the literature. The main N-containing compounds
detected by PTR-ToF-MS are isocyanic acid (HNCO), nitrous acid (HONO), hydrogen cyanide
(HCN), and ammonia ($NH_3$). HNCO, HONO, and HCN have a high contribution of the high-
temperature factor ($F_{high-T}$ = 80-100% in Figure 5a), while $NH_3$ falls into the category with a
large contribution from the low-temperature factor ($F_{low-T}$ = 86% in Figure 5e). Nitrogen in
biomass typically exists as amino acids/proteins and pyrrole/pyridine (aromatic N-heterocycles).
During the pyrolysis of those N-functionalities at high temperature (700-1100 °C), HCN is
identified as the main product in most cases (Johnson and Kang, 1971; Haidar et al., 1981;
Patterson et al., 1968; Houser et al., 1980). $NH_3$, resulting from the lower-temperature pyrolysis
of proteins, has been classified as smoldering combustion gases and falls here into the low
temperature profile (Yokelson et al., 1996).

The present analysis predominantly focuses on VOCs. The VOC emissions from biomass

burning are dominated by pyrolysis reactions of biopolymers. However, not all species are
emitted from pyrolysis reactions. For example, flaming combustion releases $CO_2$, $NO_x$, HONO,
and black carbon, etc. This is a separate process and cannot be expected to be captured by our
VOC framework. In Section 3.5 we show that MCE, which delineates flaming versus smoldering
combustion, is a poorer descriptor of VOC variability than the high versus low-temperature
pyrolysis framework.

**3.3 Chemical characteristics of VOC emissions depending on pyrolysis temperature**

**3.3.1 VOC composition**

The VOC emission profiles for the high- and low-temperature factors are shown in Figure 3 and they mainly consist of hydrocarbons, oxygenates with $n=1$-$7$ oxygen atoms, and nitrogen- and/or sulfur-containing hydrocarbons (Figure 6). In each emission profile, about half of the fraction (in ppbv) is accounted for by a combination of the following seven compounds: (i) ethene ($C_2H_4$), (ii) formaldehyde (HCHO), (iii) methanol ($CH_3OH$), (iv) acetaldehyde ($CH_3CHO$), (v) acrolein ($CH_2=CHCHO$), (vi) acetic acid ($CH_3COOH$) and glycolaldehyde ($HOCH_2CHO$), and (vii) ammonia ($NH_3$). The other half includes several fundamental structures, with a variety of functionalities, as discussed later. Oxygenates with one oxygen are predominant in both emission profiles, accounting for 39% of molar emissions in the high-temperature profile and 36% in the low-temperature profile. Emissions of highly oxygenated compounds ($\geq 2$ oxygen atoms) and ammonia are higher in the low-temperature profile than in the high-temperature profile. The fractions of hydrocarbons and compounds that contain both N and O, such as HNCO, are lower in the low-temperature profile.

VOCs emitted from biomass burning can be generally organized into major structural groups: furans, aromatics, oxygenated aromatics, aliphatic compounds, and so on. Within each structural category, compounds can have various functionalities, such as alcohol or alkene substituents (Hatch et al., 2015). VOC composition classified by 11 structures and 17 functionalities is shown in Figures 7 and 8. Some VOCs have multiple functional groups. These are counted once in each relevant category. For example, guaiacol is counted in "Oxygenated aromatic" structural category as "Alcohol" and "Ether (methoxy)" functional groups.

The most dominant emissions are attributable to aliphatic oxygenates, i.e., 62% of molar emissions in the high-temperature profile and 60% in the low-temperature profile (Figure 7). This is due to the specific compounds (ii)-(vi) described above. The low-temperature profile is twice as rich in aromatic oxygenates ($\geq 2$ oxygen atoms) and furans as the high-temperature profile, while the high-temperature profile is enriched in aliphatic (mostly alkenes) and aromatic hydrocarbons. Terpenes (including isoprene, monoterpenes, sesquiterpenes, and oxygenated monoterpenes) emitted from distillation, not from pyrolysis, are dominantly grouped with the high-temperature factor. Compared to the low-temperature profile, the high-temperature profile is enriched in the following functional groups: C-C double bond (>C=C<), C-C triple bond (-

C≡C-), diene (>C=C-C=C<), polycyclic aromatic hydrocarbon (PAH), nitrile (-C≡N), amide (-
C(=O)-N-), nitro (-NO$_2$), nitrate (-NO$_3$), thiol/sulfide (-S-(H)) (Figure 8). The low-temperature
profile is enriched in alcohols (-OH), ethers (mostly methoxy groups: -O-CH$_3$), esters (-C(=O)-
O-), and amines (-NH$_2$; mostly ammonia). The emissions of compounds with carbonyl groups
(>C=O) and acids (-C(=O)-OH-) are similar. These results are consistent with the contributions
of VOC to the high- and low-temperature factors described in Section 3.2.

### 3.3.2 OH reactivity

The hydroxyl radical (OH) is an important driver of daytime oxidation chemistry.
Quantifying the VOC reactivity with OH provides insight into which VOC emissions may be
most important for ozone and secondary organic aerosol (SOA) formation. Interestingly, the two
profiles have a similar average per-molecule (weighted by abundance) rate constant with OH:
$15.7 \times 10^{-12}$ cm$^3$/molecule/s for the high-temperature profile and $15.8 \times 10^{-12}$ cm$^3$/molecule/s for
the low-temperature profile. However, the reactivity is provided by very different VOCs in each
profile. Aliphatic oxygenates are important in both profiles, but more so in the high-temperature
profile (30% of reactivity) than in the low-temperature profile (24% of reactivity). In the high-
temperature profile, the reactivity also has a large contribution from terpenes and aliphatic
hydrocarbons, while in the low-temperature profile, the reactivity is largely due to furans and
aromatics (Figure 9a). Since the total VOC emissions in real-world fires come from a mixture of
the high- and low- temperature pyrolysis factors, the total OH reactivity of fresh emissions
should scale directly with VOC concentration.

### 3.3.3 Volatility

Volatility is another important chemical characteristic affecting secondary organic aerosol
(SOA) yield and formation rate. The low-temperature emission profile contains more compounds
that are of higher molecular weight, more oxygenated, and of lower volatility (Figure 9b).
Oxygenated aromatics have been shown to be important biomass burning SOA precursors (Bruns
et al., 2016), and while the SOA yields of many other compounds are unknown, the lower
volatility and higher oxygen content of the low-temperature profile suggests a potentially more
efficient SOA formation. SOA formation was also studied during the FIREX 2016 campaign, by
oxidizing emissions in a chamber, and will be presented separately (Lim et al, in prep, 2018). We
note that the compounds with $C_0 < 10^2$ µg m$^{-3}$ shown in Figure 9b should be primarily in the
particle phase and not measureable by PTR-MS without long delay times (Pagonis et al., 2017).
However, the volatility of these compounds (calculated from the elemental composition) has an
uncertainty of several orders of magnitude. Also, the cyclic compounds that are abundant in the
low-temperature profile, such as aromatic oxygenates, produce multifunctional ring-opening-
products that are known to be efficient SOA precursors (Yee et al., 2013). In a similar manner to
the OH reactivity, the total volatility distribution can be estimated from the relative importance
of the high- and low-temperature pyrolysis in a given fire.

**3.4 Relationship of fuel characteristics to relative importance of high- and low-temperature**

**pyrolysis factors**
To use the PMF profiles (Figure 3) for estimates of VOC emissions from other fires, it is
necessary to know the relative fire-integrated contributions of high- and low-temperature
pyrolysis for those fires. As a step in this direction, in the present work, we found that fire-
integrated molar emission ratios of total VOCs from high-temperature pyrolysis to low-
temperature pyrolysis, $\sum VOC_{high-T}$ (in ppbv)/$\sum VOC_{low-T}$ (in ppbv), are related to which parts of
the plants are burned (blue bars in Figure 10). When leafy fuels (i.e., canopy, shrub, and
herbaceous fuels) are burned, the fraction of total VOC emissions originating from high-
temperature pyrolysis is higher than those from low-temperature pyrolysis. These results imply
that surface-to-volume ratios and the content of biopolymers in a given fuel can strongly affect
the relative importance of high- and low-temperature pyrolysis. Leaves have high surface-to-
volume ratios and despite higher fuel moisture, at least the surface may tend to heat up easily,
resulting in a higher contribution from the high-temperature factor. The higher monoterpene
content of foliage may explain why low-temperature distillation products like monoterpenes are
associated with the high-T pyrolysis factor.
In contrast, the burn of rotten wood was found to contain VOC emissions from low-
temperature pyrolysis only. Our brown rotten wood samples were enriched in lignin (Kirk and
Cowling, 1984). Lignin is relatively resistant to thermal decomposition compared to cellulose
and hemicellulose. The temperature range where pyrolytic decomposition occurs significantly is
280-500 °C for lignin, 240-350 °C for cellulose, and 200-260 °C for hemicellulose (Liu et al.,
2016; Babu, 2008), as shown in Figure 4b. In our laboratory fires, the rotten wood first
smoldered for an extended period, and then flames were observed. However, only the low-
temperature profile was observed. This suggests that it is more difficult for lignin-rich fuels to
reach high enough temperatures to emit the "high-temperature pyrolysis" VOCs. Therefore, we
do not see the same gradient in pyrolysis products that is observed for other fuel burns mainly
consisting of cellulose and hemicellulose. Nitrogen content and speciation also vary between
different biomass components, and temperature and differences in biopolymer content have been
shown to strongly affect the composition of nitrogen-containing emissions (Hansson et al., 2004;
Ren et al., 2011; Coggon et al. 2016). This is consistent with the observed differences in nitrogen
speciation between the two profiles.

**3.5 High- and low-temperature pyrolysis profiles describe total VOC emissions**

Previous studies have found a correlation between the emission factors of certain VOCs and
the fire-integrated modified combustion efficiency (MCE) (Yokelson et al., 1996: Yokelson et al.,
1997; Selimovic et al., 2018). Thus, one might expect that the high- and low-temperature
pyrolysis factors would also show a strong relationship to MCE. However, MCE does not
parameterize the relative amounts of high- and low-temperature pyrolysis products very well,
either instantaneously or on a fire-integrated basis (Figure 11). The basic reason is that $CO_2$ as
well as $NO_x$ are emitted overwhelmingly from flaming combustion, which is *not* the main source
of most VOC emissions, and these emissions are not expected to correlate with a linear
combination of the high- and low-temperature pyrolysis processes, while CO emissions are
reasonably well correlated with an average of high- and low-temperature emissions (Figures 1
and S6). This is especially clear in rotten log burns, where $CO_2$ and the PMF profiles are not
correlated. The $CO_2$ emissions are enhanced by shifting from the smoldering to flaming
combustion, but VOC emission patterns are not changed from the low- to high-temperature
pyrolysis (Figure S7). Consequently, $CO_2$ and MCE, which indicate the separation between
flaming and smoldering combustions, are not appropriate to estimate the high-/low-temperature
pyrolysis VOC emissions. Our results indicate that VOC emissions are even more closely
correlated to the biopolymer composition and the surface-to-volume ratios of fuels, than to the
MCE. It is also seen that for some fires the air temperature correlates with the high-temperature
contribution (e.g., Fires #37 and #59 shown in Figure S8a-c). This suggests that the VOC
emissions are certainly related to the temperature within a fire. However, some other burns did
not have a good correlation between the temperature and VOC emissions (e.g., Fire #38 shown
in Figure S8d), because the temperature measurement had some issues in the present work: (i)
background temperature for each burn was different, (ii) some burns have colder temperature at
end compared to start, which means that the laboratory was not controlled at constant
temperature, and (iii) the increase in air temperature often lagged behind the emissions,
especially at the start of a fire.
The relative contributions from the high- and low-temperature processes could be estimated
from ratios of distinct marker species that are consistently enhanced in the high and low-
temperature profiles. Several such pairs were considered and the ratio of ethyne ($C_2H_2$) to furan
($C_4H_4O$) can reasonably predict the ratio of high- to low-temperature emissions as given in Eq. 1:

$$\frac{total\ VOC\ ,high\ temperature\ (ppbv)}{total\ VOC\ ,low\ temperature\ (ppbv)} = \frac{ethyne\ (ppbv)\ /\ 0.0393}{furan\ (ppbv)\ /\ 0.0159} \qquad (1)$$


The derivation and how the ethyne/furan ratio correlates with the high-/low-temperature
emission ratio are given in the Supporting Information (S2 and Figure S9). However, this pair is
not ideal because measurements of these two species are not frequently available and furan has
high reactivity to both $O_3$ and $NO_3$ radicals. Future work should assess non-PTR measurements
in order to find appropriate external markers.
Studies of laboratory burns and wildfires have reported variable emission ratios (or factors)
for various VOCs as well as fire-integrated MCE, even for similar fuel types. Here we
investigate how well total VOC emissions in biomass burning can be fit by the average VOC
emission profiles (Figure 3) using emission factors and ratios reported in the literature for
laboratory and field burns (Gilman et al., 2015; Stockwell et al., 2015; Akagi et al., 2011). When
fitting the present high- and low-temperature factors to the other biomass burning data, total
VOC emissions can be described with different relative fractions of the factors (Figure S10). For
example, the best fit to a laboratory study by Gilman et al. (2015), using fuels from southwestern,
southeastern, and northern U.S. (e.g., pine, spruce, fir, chaparral, mesquite, and oak) with MCE =
0.75-0.98, includes 32% high-temperature and 68% low-temperature VOC emissions; for
another laboratory study by Stockwell et al. (2015) including several types of grass, spruce, and
chaparral with MCE = 0.68-0.99, 59% high temperature and 41% low temperature; temperate
forest fires (MCE = 0.95) reported by Akagi et al. (2011), 77% high temperature and 23% low
temperature, while in the case of chaparral fires (MCE = 0.96), 48% high temperature and 52%
low temperature. The fitting can be done with high correlation coefficient ($r \geq 0.92$) for all the
literature data (Figure S10). This is further evidence that at most two factors can explain the
majority of VOC variability. Therefore, these two factors could be used to fill in VOCs not
measured in the other studies which sometimes had less chemical detail. The current study
incorporated a wide range of MCEs and fuel moisture contents (Table 1), so the two-factor
description may be applicable under many conditions. However, some other factors should be
required for specific burns, as discussed below.
**3.6 Emission of specific compounds**
**3.6.1 Distillation phase**
At the beginning of many burn experiments, a white smoke is visible immediately prior to
ignition. This "distillation phase" does not result from pyrolysis or combustion, but rather a
gradual heating and release of water and volatile compounds trapped within the biomass. This
phase of the fire was not distinguished by PMF. The distillation phase from coniferous fuels is
enriched in some compounds highly relevant to atmospheric chemistry, especially terpenes (Koss
et al., 2018). But this phase lasts only a short time (typically less than 10 seconds), in which only
a short spike in emissions is observed. Accordingly, PMF cannot capture this phase effectively
even if a large number of factors is chosen. As an exception, the distillation phase of Sagebrush,
enriched in terpenes and a specific oxygenated monoterpene (camphor), can be distinguished as a
third PMF factor, because that phase lasted longer than 30 seconds in that fire. The reported
overall residual of 15% includes the poorly fitted distillation phase, and we stress that it typically
accounts for only a small portion of the overall emissions. Additionally, with the exception of
terpenes, the composition of the distillation profile is similar to that of the high-temperature
profile.
For some fuel burns other than coniferous fuels (e.g., Manzanita), VOC emissions during the
distillation phase are quite small, although distillation smoke is visible. In these cases, PMF
incorporates this phase into the low-temperature pyrolysis factor. There may be a relationship
between the VOC emission process coincident with distillation (low- or high-temperature) and
the presence of visible smoke. For instance, perhaps here the temperatures are low enough that
the compounds are able to re-condense into visible smoke.

**3.6.2 Duff burn**

A fourth factor can be resolved from the PMF analysis of duff burns. The distribution of VOC structures and functionality in the duff emission profiles (Figure 12a) is similar to the low-temperature pyrolysis profile (Figure 12b). The major difference is much higher emission of aliphatic nitrogen-containing compounds: 56% more of these compounds are emitted per-ppbv VOC in the duff profile than in the low-temperature profile. The additional emissions are mostly nitriles and amides, especially hydrogen cyanide (HCN), acetonitrile, and acetamide. Pyrroles and pyridines are also enhanced, but are much less abundant overall.

The organic portion of duff is enriched in nitrogen relative to other components of coniferous fuels. The nitrogen to carbon ratio in the Subalpine fir duff (N:C ratio = 0.028 by weight) was a factor of 2.1 higher than the average of other Subalpine fir components, and the Engelmann spruce duff N:C ratio (0.022) was 1.3 times higher than other Engelmann spruce components. Coggon et al. (2016), who investigated VOC emissions from the burning of herbaceous and arboraceous fuels, also found that the nitrogen-containing fraction of VOCs emitted from biomass burning increased with the nitrogen content of the fuel.

However, the nitrogen content cannot entirely explain why duff has a unique emission profile. Other fuels, such as Ceanothus and Ponderosa pine litter, have similar N:C ratios (0.025, 0.024, and 0.022, respectively) but are explained well by the 2-factor PMF solution consisting of high- and low-temperature pyrolysis factors. The contradiction may be due to differences in the speciation of nitrogen-containing organics. In woody and leafy fuels, proteins and amino acids account for 80-85% of the organic nitrogen (Ren and Zhao, 2015). In soils, proteins account for typically only 40% of organic nitrogen, and heterocyclic nitrogen compounds (pyrroles and pyridines) account for 35% (Schulten and Schnitzer, 1997). Pyrolysis of nitrogen heterocycles releases HCN, while proteins and amino acids may release more $NH_3$ (Leppälahti and Koljonen, 1995). This is consistent with the higher HCN and nitriles characteristic of the duff emission profile.

**3.6.3 Variation in specific VOCs between fuels**

When comparing emission profiles of individual fuels to the average profiles shown in
Figure 3, there are some specific compounds whose emissions are notably higher ($> \times 5$) or lower
($< \times 0.2$) than the average (Figure S4). Here we highlight several key features:
(i)   For Ponderosa/Lodgepole/Loblolly pines, Douglas/Subalpine firs, and Juniper, the
emission of benzoquinone ($C_6H_4O_2 \cdot H^+$, $m/z$ 109.028) is quite low in the high-
temperature pyrolysis: 7-21% of the average emission for the pines and firs, and 2%
for Juniper (Figures S4a-1~4, 6, and 7).

(ii)  For fuels other than coniferous fuels and Sagebrush, i.e., Bear grass, Excelsior,
Ceanothus, Chamise, and Manzanita, emissions of monoterpenes ($C_{10}H_{16} \cdot H^+$, $m/z$
137.132) are only 2-15% of the average (Figures S4a-8~14).

(iii) Excelsior emits especially low quantities of nitrogen-containing compounds,
especially nitriles (hydrogen cyanide, acetonitrile, acrylonitrile, and propane nitrile)
and pyridine, in the high-temperature pyrolysis (Figure S4a-9). This is because the
nitrogen content in Excelsior is significantly lower than other fuels. The Excelsior
600            N:C ratio (0.005 by weight) is 3.6 times lower than the average of other fuels (0.017
$\pm$ 0.006).

(iv)  High-temperature pyrolysis of Ceanothus produces quite high emission of
benzofuran-type compounds (Figure S4a-10). Benzofuran ($C_8H_6O \cdot H^+$, $m/z$ 119.049)
and methylbenzofuran and possibly its isomer such as cinnamaldehyde ($C_9H_8O \cdot H^+$,
$m/z$ 133.065) are 5.5 and 10.1 times higher than the average, respectively.

(v)   Sagebrush specifically emits camphor ($C_{10}H_{16}O \cdot H^+$, $m/z$ 153.127) in high-
temperature pyrolysis (Figure S4a-15).

(vi)  There are a limited number of exceptions in low-temperature profiles (Figure S4b).
This means that low-temperature pyrolysis gives almost identical VOC emissions,
independent of fuel types.


**4 Conclusions**
This work focused on interpretation of VOC emissions from biomass burning. We provided
an understanding of VOC variability based on known chemical and physical processes to release
VOCs from fires. We explained most of the observed variability between VOC emissions from
fuel types and over the course of a fire using just two emission profiles: (i) high-temperature
pyrolysis profile and (ii) low-temperature pyrolysis profile. The results are summarized as
follows:

1. The two profiles can explain the variability in VOC emissions composition between

different fuel types and over the course of individual fires, with an average residual of <

15%.

2. The high-temperature profile is quantitatively similar between different fuel types ($r^2$ >

0.84), and likewise for the low-temperature profile.

3. The two profiles are significantly different in terms of VOC composition, volatility, and

contributors to OH reactivity. The high-temperature pyrolysis profile is enriched in

aliphatic unsaturated hydrocarbons, (polycyclic) aromatic hydrocarbons, terpenes

(emitted from distillation), HCN, HNCO, and HONO. The resulting OH reactivity is

primarily attributed to terpenes, aliphatic hydrocarbons, and non-aromatic oxygenates.

The low-temperature pyrolysis profile is enriched in aromatic oxygenates, furans, and

$NH_3$. The OH reactivity is contributed significantly by furans and aromatics.

4. The fire-integrated molar emission ratios of total VOCs from high-temperature

pyrolysis to low-temperature pyrolysis are related to the biopolymer composition and

surface-to-volume ratios of fuels. Higher surface-to-volume ratios lead to total VOC

emissions enriched in products resulting from high-temperature pyrolysis than from

those resulting from low-temperature pyrolysis.

5. The two VOC profiles can model previously reported VOC data for laboratory and field

burns ($r \geq 0.92$). This suggests that these two profiles could be used to fill in VOCs not

actually measured in the previous studies which sometimes had less chemical detail.

6. MCE, which parameterizes flaming and smoldering combustion, is not appropriate to

estimate the high-/low-temperature pyrolysis VOC emissions. This suggests that the

high- and low-temperature pyrolysis profiles may provide information on emissions that

is not accessible with a broader definition of smoldering combustion implicit in the use

of MCE.

7. Duff burns emit a specific VOC profile which is similar to that of low-temperature

pyrolysis, but additionally includes aliphatic nitrogen-containing compounds, especially

HCN, acetonitrile, and acetamide.

Our framework provides a way to understand VOC emissions variability in other laboratory and
field studies of biomass burning. We highlight two areas of useful future work. First, external
tracers should be found that will allow the prediction of the relative contribution of individual
profiles. This could include specific chemical species, an understanding of how fuel or burn
characteristics relate to the relative contribution of the two profiles, or a relationship between
some measure of fire temperature and the VOC profiles. Second, the SOA and ozone formation
potential of the two profiles should be determined. With this further work, the VOC profiles
could be widely useful to model VOC emissions from many types of biomass burning in the
western US, with additions to the framework being needed for fires that burn a lot of duff.
Future work should also include a quantitative comparison of the VOC PMF results to
measurements of aerosol, inorganic gases, and organic species not measured by PTR-ToF-MS.
Such a comparison would help define the relationship between VOCs and characteristics of
primary organic aerosol (POA). We note that the primary aerosols have also been shown to have
distinct profiles that correlate with different pyrolysis and combustion processes in the fire
(Reece et al., 2017; Haslett et al., 2017).


**Acknowledgments**
K. Sekimoto acknowledges the Postdoctoral Fellowships for Research Abroad from Japan
Society for the Promotion of Science (JSPS) and a Grant-in-Aid for Young Scientists (B)
(15K16117) from the Ministry of Education, Culture, Sports, Science and Technology of Japan.
A. Koss acknowledges support from the NSF Graduate Fellowship Program. M. Coggon
acknowledges the Visiting Postdoctoral Fellowship from Cooperative Institute for Research in
Environmental Sciences (CIRES). V. Selimovic and R. J. Yokelson were supported by NOAA-
CPO grant NA16OAR4310100. J. de Gouw worked as a consultant for Aerodyne Research Inc.
during part of the preparation phase of this manuscript. We thank for support from NOAA AC4
external funding, and thank the USFS Missoula Fire Sciences Laboratory for their assistance and
cooperation. This work was also supported in part by NOAA's Climate Change and Health of the
Atmosphere initiatives.

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

**Table 1. (a) Data numbers and corresponding details of 15 different fuels used in PMF analysis. (b) Average MCE and fuel**

**moisture content. (c) Residuals of 2-factor PMF solutions. (d) Correlation with average VOC emission profile (Figure 3).**

| Fuel | ( a ) Data number for consolidated PMF | | | ( b ) Fire characteristics | | ( c ) Residual [%] [a] | ( d ) Correlation with average VOC emission profile (Figure 3) | | | |
|---|---|---|---|---|---|---|---|---|---|---|
| | | | | | | | High-temperature pyrolysis factor | | Low-temperature pyrolysis factor | |
| | Total | Detail | | MCE | Moiture content | Average ± STDV (maximum, minimum) | Slope | Correlation coefficient ($r^2$) | Slope | Correlation coefficient ($r^2$) |
| 1. Ponderosa pine | 10 | Realistic 5 (Fire 01, 02, 37, 59, 72) | | 0.913-0.940 | 24.3-31.8% | 15.7 ± 7.6 (28.9, 7.7) | 0.976 ± 0.004 | 0.9393 | 1.012 ± 0.005 | 0.9245 |
| | | Canopy 2 (Fire 19, 39) | | 0.904-0.935 | 40.4-51.1% | | | | | |
| | | Litter 1 (Fire 38) | | 0.945 | 6.2% | | | | | |
| | | Rotten log 2 (Fire 13, 73) | | 0.932-0.957 | 2.9-5.7% | | | | | |
| 2. Lodgepole pine | 7 | Realistic 4 (Fire 06, 07, 58, 63) | | 0.927-0.943 | 20.3-24.4% | 14.8 ± 4.9 (23.3, 10.9) | 0.990 ± 0.004 | 0.9586 | 0.990 ± 0.002 | 0.9716 |
| | | Canopy 1 (Fire 40) | | 0.924 | 49.30% | | | | | |
| | | Litter 2 (Fire 21, 41) | | 0.925-0.938 | 7.0-10.5% | | | | | |
| 3. Loblolly pine | 2 | Litter 2 (Fire 35, 53) | | 0.922-0.929 | 5.4-10.9% | 6.3 ± 0.3 (6.6, 6.1) | 0.989 ± 0.007 | 0.8662 | 0.960 ± 0.004 | 0.8862 |
| 4. Douglas fir | 4 | Realistic 2 (Fire 14, 57) | | 0.926-0.951 | 23.3-25.7% | 21.2 ± 9.3 (34.9, 14.9) | 0.996 ± 0.004 | 0.9508 | 0.999 ± 0.003 | 0.9563 |
| | | Canopy 1 (Fire 18) | | 0.928 | 50.3% | | | | | |
| | | Litter 1 (Fire 43) | | 0.951 | 3.0% | | | | | |
| 5. Engelmann spruce | 3 | Realistic 1 (Fire 08) | | 0.920 | 13.0% | 20.5 ± 2.4 (22.2, 18.8) [b] | 0.999 ± 0.006 [b] | 0.9019 [b] | 0.960 ± 0.004 [b] | 0.9004 [b] |
| | | Canopy 1 (Fire 25) | | 0.950 | 34.0% | | | | | |
| | | Duff 1 (Fire 26) | | 0.817 | 0.6% | 82.0 | - | - | - | - |
| 6. Subalpine fir | 6 | Realistic 2 (Fire 47, 67) | | 0.932-0.942 | 32.8-35.6% | 23.0 ± 14.1 (45.2, 9.1) [b] | 1.001 ± 0.005 [b] | 0.9359 [b] | 0.999 ± 0.003 [b] | 0.9547 [b] |
| | | Canopy 2 (Fire 15, 23) | | 0.886-0.947 | 17.6-55.5% | | | | | |
| | | Litter 1 (Fire 51) | | 0.906 | 6.6% | | | | | |
| | | Duff 1 (Fire 56) | | 0.886 | 0.9% | 87.0 | - | - | - | - |
| 7. Juniper | 2 | Canopy 2 (Fire 68, 75) | | 0.928-0.939 | 45.0-48.0% | 6.4 ± 3.0 (8.5, 4.3) | 1.016 ± 0.006 | 0.8872 | 0.971 ± 0.004 | 0.9010 |
| 8. Bear grass | 1 | - (Fire 62) | | 0.897 | 55.1% | 5.6 | 1.039 ± 0.006 | 0.8847 | 1.006 ± 0.004 | 0.9174 |
| 9. Excelsior | 2 | - (Fire 49, 61) | | 0.945-0.971 | 3.9-5.4% | 6.2 ± 3.0 (8.3, 4.0) | 1.04 ± 0.01 | 0.6806 | 1.012 ± 0.007 | 0.8521 |
| 10. Ceatnothus | 2 | Shrub 2 (Fire 69, 74) | | 0.942-0.947 | 17.7-27.9% | 10.2 ± 1.1 (11.0, 9.5) | 1.000 ± 0.007 | 0.8416 | 1.030 ± 0.006 | 0.8985 |
| 11. Chamise (contaminated) | 3 | Canopy 3 (Fire 24, 29, 46) | | 0.948-0.959 | 10.9-16.1% | 13.1 ± 3.9 (16.0, 8.6) | 1.037 ± 0.006 | 0.8951 | 1.044 ± 0.004 | 0.9477 |
| 12. Chamise (uncontaminated) | 3 | Canopy 3 (Fire 27, 32, 48) | | 0.946-0.954 | 6.2-17.1% | 12.6 ± 2.0 (14.2, 10.4) | 1.017 ± 0.005 | 0.9322 | 1.024 ± 0.004 | 0.9299 |
| 13. Manzanita (contaminated) | 2 | Canopy 2 (Fire 30, 33) | | 0.962-0.963 | 23.5-26.7% | 13.0 ± 1.1 (13.8, 12.3) | 0.997 ± 0.004 | 0.9347 | 1.034 ± 0.004 | 0.9504 |
| 14. Manzanita (uncontaminated) | 2 | Canopy 2 (Fire 28, 34) | | 0.963-0.964 | 25.7-26.3% | 7.3 ± 1.1 (8.0, 6.5) | 1.015 ± 0.005 | 0.9229 | 1.043 ± 0.005 | 0.9224 |
| 15. Sagebrush | 2 | Shrub 2 (Fire 66, 71) | | 0.919-0.922 | 37.8-54.2% | 7.0 ± 2.1 (8.5, 5.6) | 0.993 ± 0.005 | 0.9046 | 1.011 ± 0.004 | 0.9306 |

[a] Residual [%] = [Total measured ion signal - Total synthetic ion signal of high- and low-temperature factors] / Total measured ion signal x 100

[b] "Duff" data is excluded.

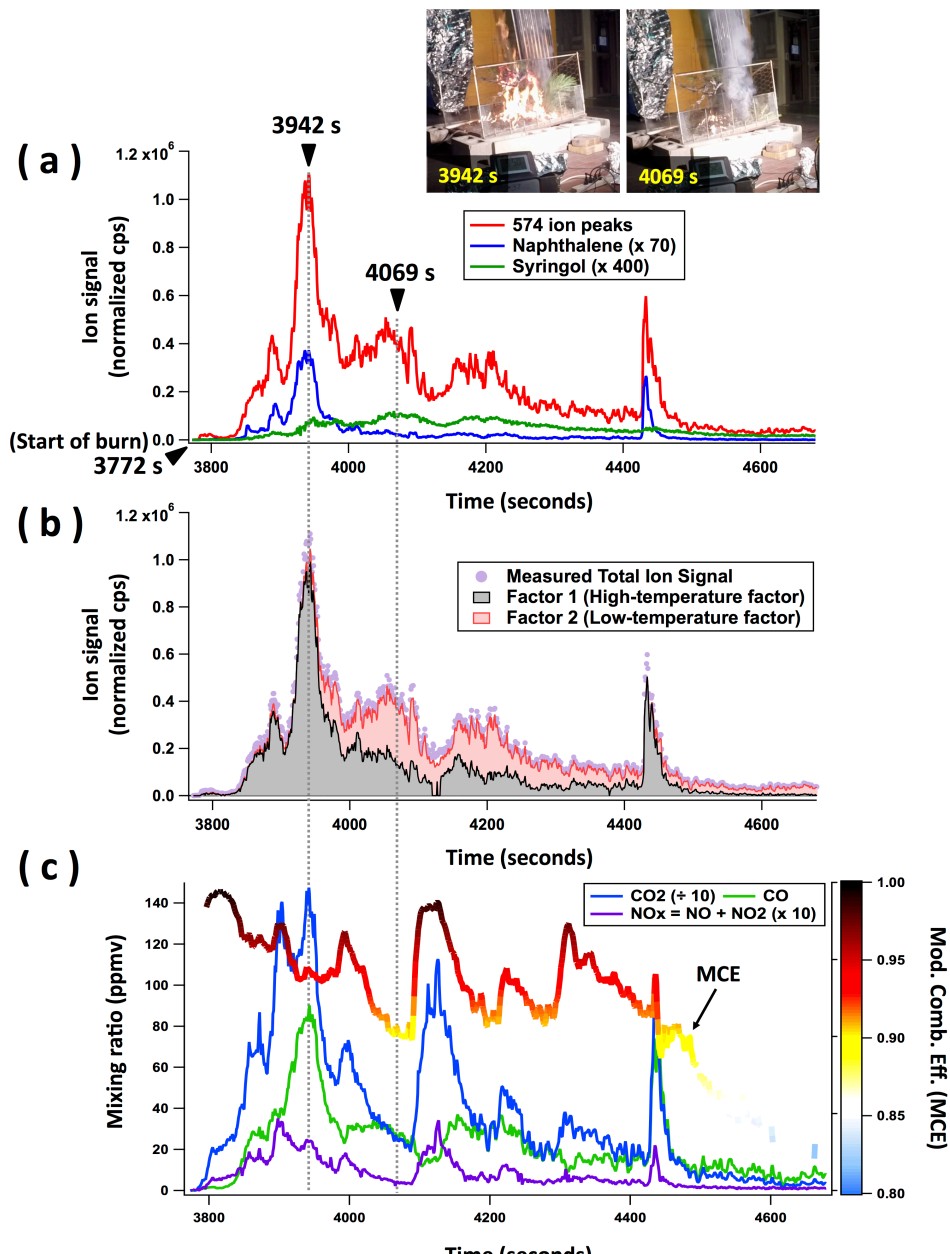


**Figure 1. Results for an example burn of Ponderosa pine realistic mixture (Fire #37). (a)**
**Time series of ion signals of 574 ion peaks, naphthalene ($C_{10}H_8 \cdot H^+$, *m/z* 129.070), and**
**syringol ($C_8H_{10}O_3 \cdot H^+$, *m/z* 155.070). (b) PMF results of 2-factor solution. The grey and pink**
**colors are stacked, not overlapped. (c) Time series of mixing ratios of $CO_2$, CO, and $NO_x$**
**measured by open-path Fourier transform infrared (OP-FTIR) optical spectroscopy and**
**the modified combustion efficiency (MCE) (Selimovic et al., 2018). The MCE trace is**
**colored by the key and scale on the right.**

**( a ) Ponderosa (Fire 72) vs. Ponderosa (Fire 02)**

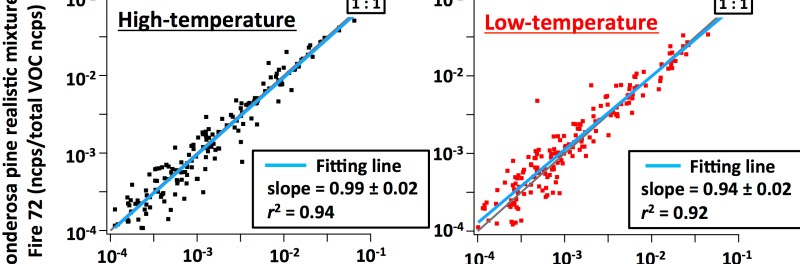

**( b ) Douglas fir vs. Ponderosa pine**

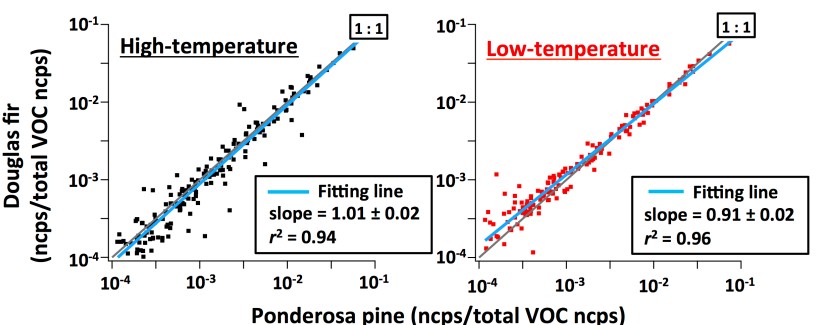

**( c ) Manzanita vs. Ponderosa pine**

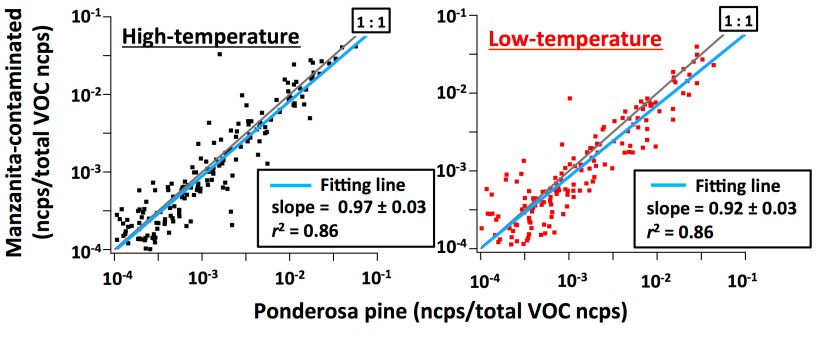

**( d ) Bear grass vs. Ponderosa pine**

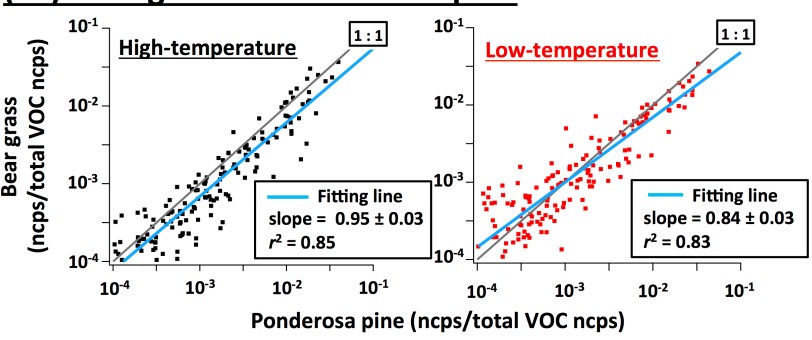

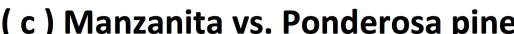

**( e ) Low-temperature vs. High-temperature pyrolysis factor**

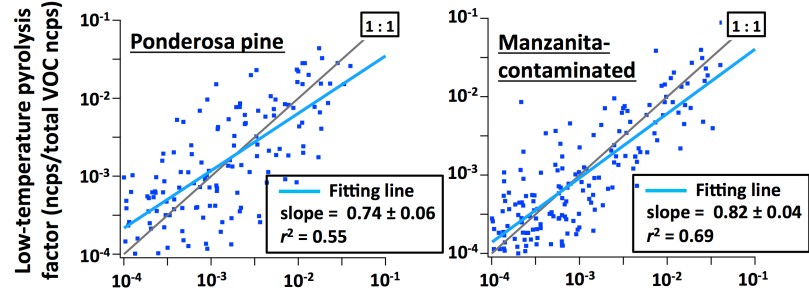

**Figure 2.** Comparison of mass spectral profiles: (a) Ponderosa pine realistic mixture (Fire #72) vs. Ponderosa pine realistic mixture (Fire #02) for high- and low-temperature pyrolysis factors. (In this case, PMF was separately performed for data of Fire #02 and #72.) (b) Douglas fir vs. Ponderosa pine for high- and low-temperature factors. (c) Manzanita (contaminated) vs. Ponderosa pine for both the factors. (d) Bear grass vs. Ponderosa pine for both the factors. (e) Low- vs. high-temperature pyrolysis factor for Ponderosa pine and Manzanita (contaminated). Data points in individual panels correspond to well-fitted 434 ion peaks. Slope and correlation coefficient ($r^2$) are obtained using logarithmic fraction, i.e., log(ncps/total VOC ncps).

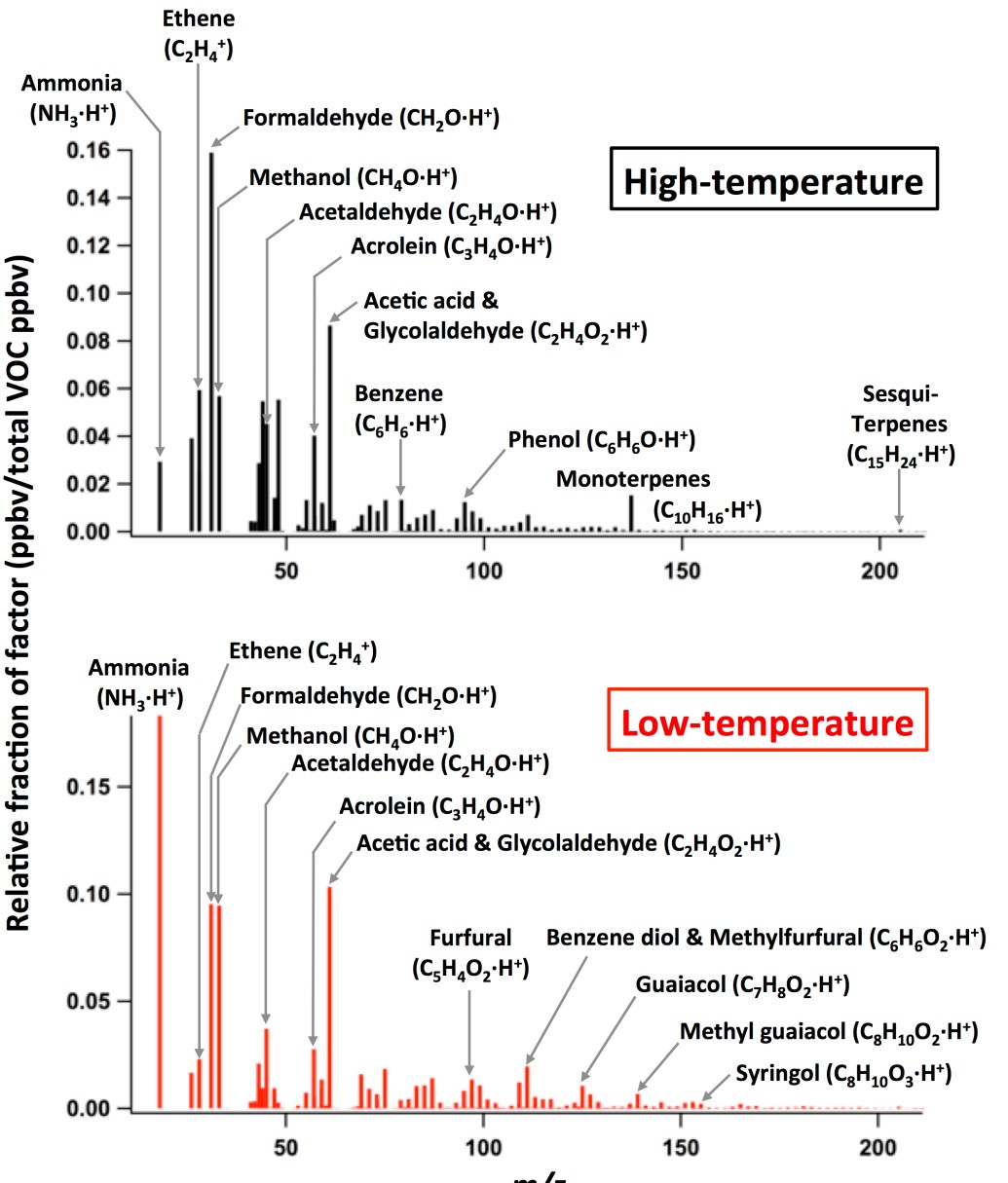

Figure 3. Average VOC emission profiles of high- and low-temperature pyrolysis factors, obtained using consolidated PMF results of 15 different fuels.

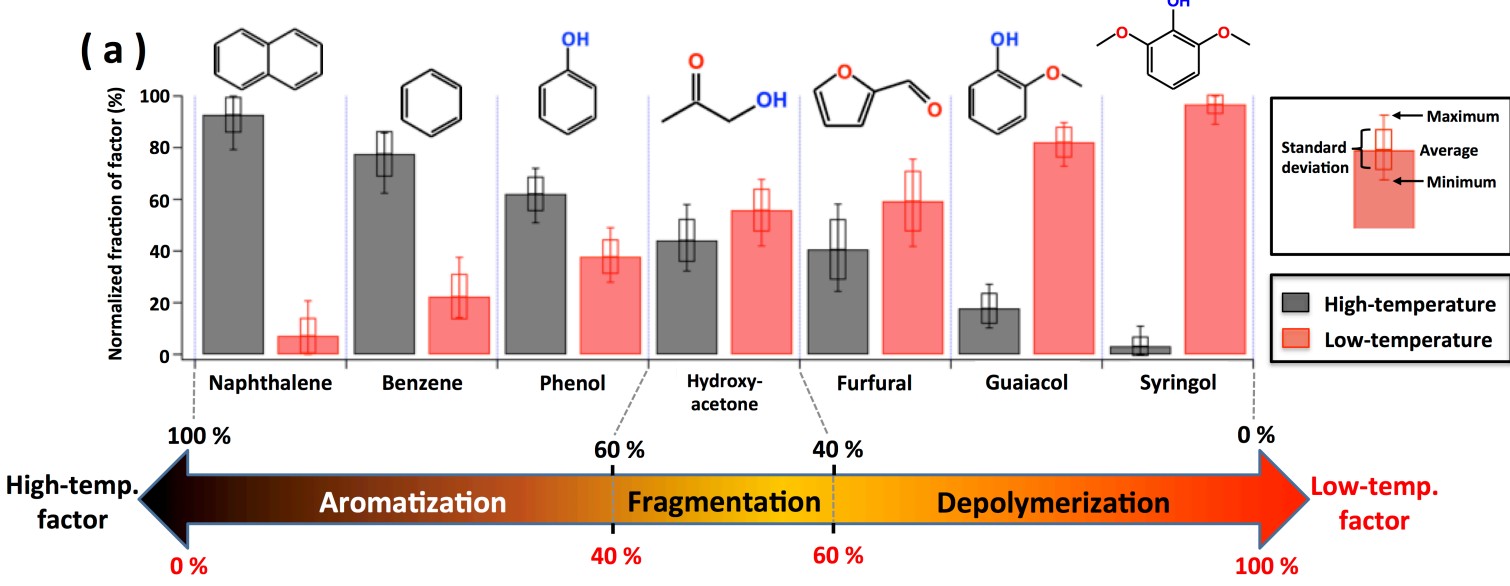

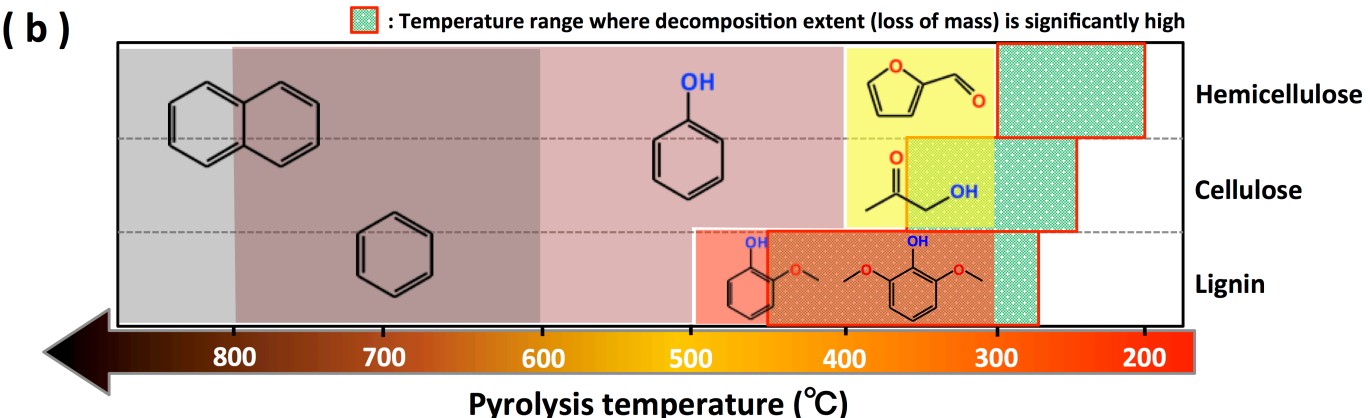

Figure 4. (a) Normalized fraction of factors for selected biomass pyrolysis products, obtained using PMF results of 15 different fuels. (b) Diagram of the relationship between pyrolysis temperature and products for hemicellulose, cellulose, and lignin, as reported in the literature (Collard and Blin, 2014). Individual color bars show the temperature range to form specific products described by chemical structures.

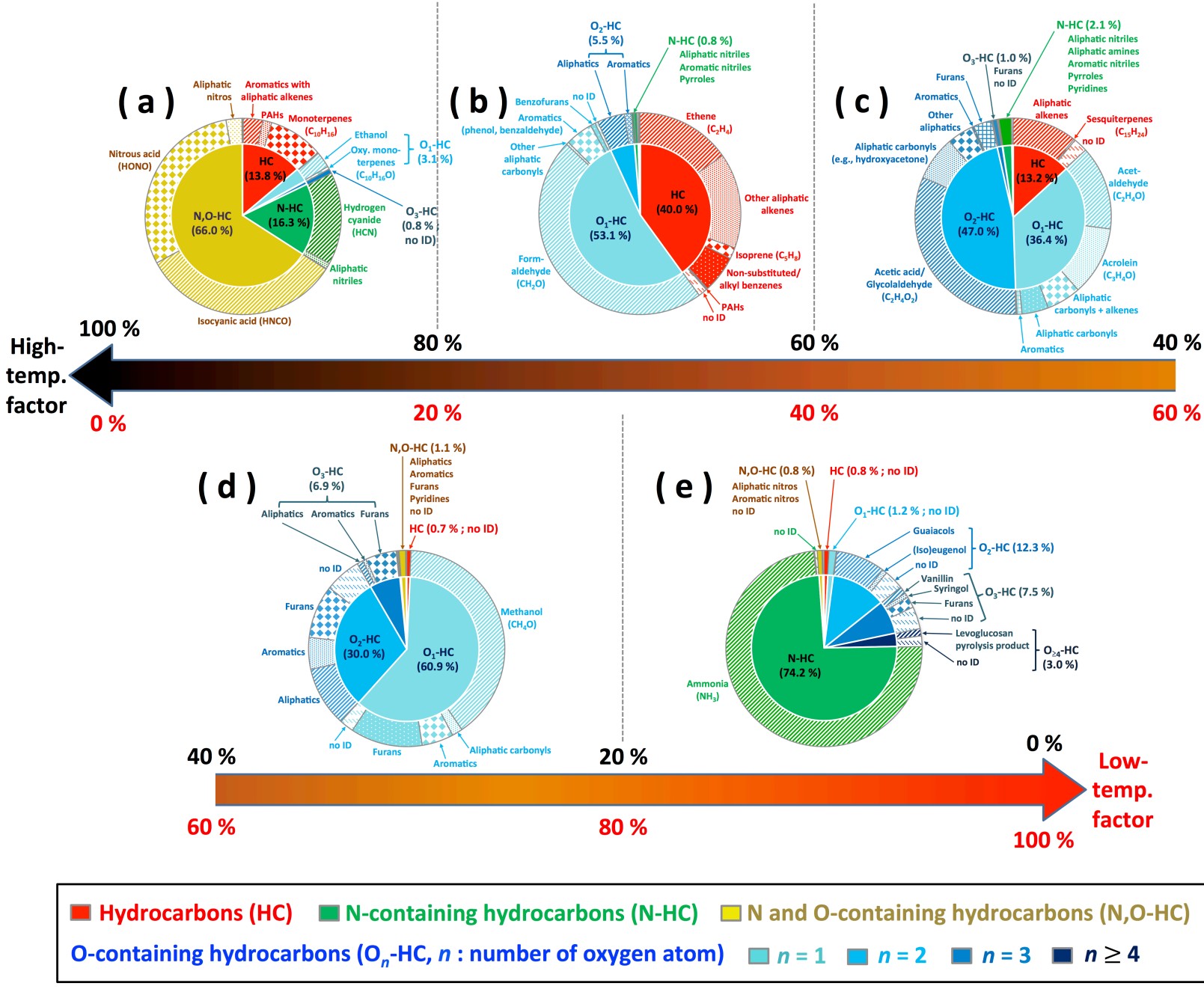

**Figure 5. Contributions, shown as normalized fractions, of VOCs relative to the high- and low-temperature factors: (a) $F_{\text{High-T}}$ = 100-80% and $F_{\text{Low-T}}$ = 0-20%, (b) $F_{\text{High-T}}$ = 80-60% and $F_{\text{Low-T}}$ = 20-40%, (c) $F_{\text{High-T}}$ = 60-40% and $F_{\text{Low-T}}$ = 40-60%, (d) $F_{\text{High-T}}$ = 40-20% and $F_{\text{Low-T}}$ = 60-80%, and (e) $F_{\text{High-T}}$ = 20-0% and $F_{\text{Low-T}}$ = 80-100%. In this figure, molar emissions (in units of ppbv) of all the ion peaks in VOC emission profiles (Figure 2b) are described. The inner circle in each pie chart shows the elemental composition of the emissions. The outer circle provides more detailed information on specific compounds, structures, and functionalities found in each group. Details of molar fractions in each category are summarized in Table S2.**

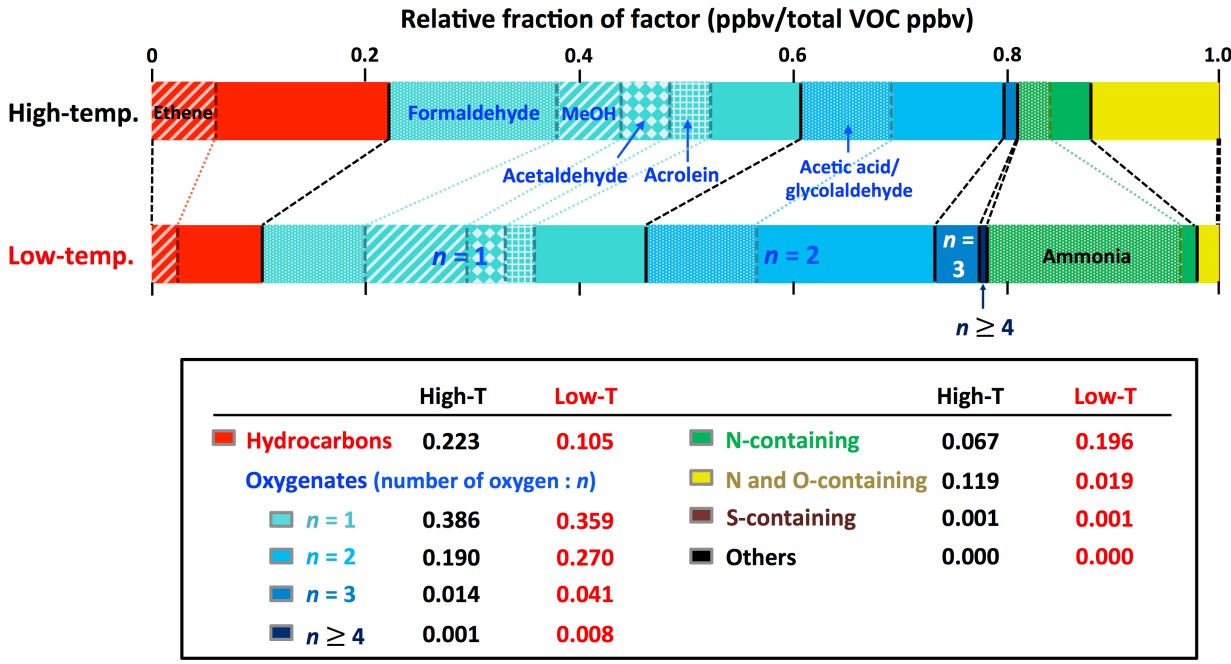

**Figure 6. VOC composition in the high- and low-temperature emission profiles.**

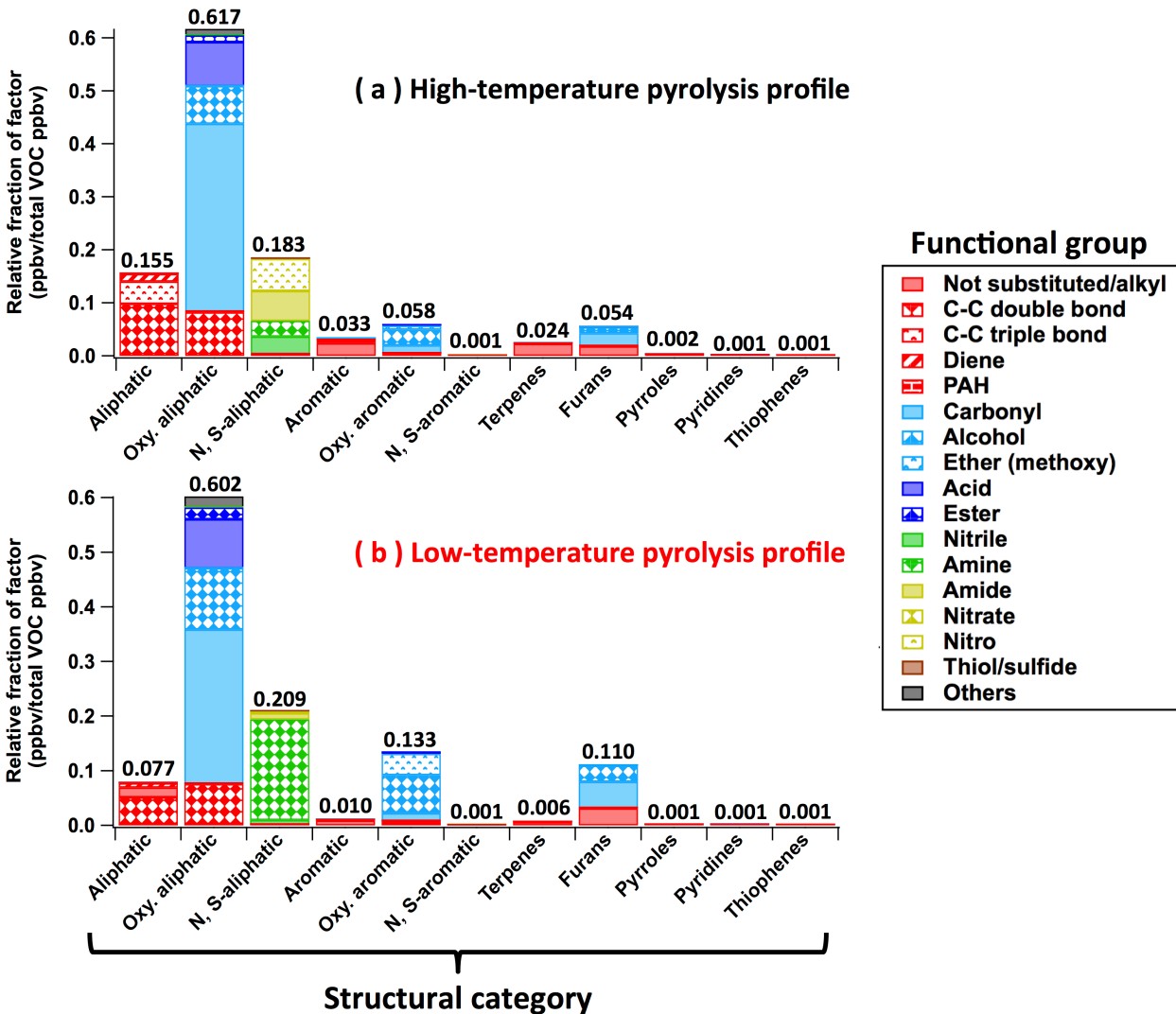

**Figure 7. VOC composition in (a) high-temperature pyrolysis and (b) low-temperature pyrolysis emission profiles (Figure 3) sorted by 11 structural categories and 17 functional groups. Some VOCs have multiple structures. These are counted once in each relevant category. For example, benzofuran is counted in the structural categories of "Oxy. aromatic" and "Furans" as "Not substituted/alkyl" functional group. Structures detected with low abundance (<0.002 ppbv/total VOC ppbv) are mostly not-substituted or alkyl-substituted.**

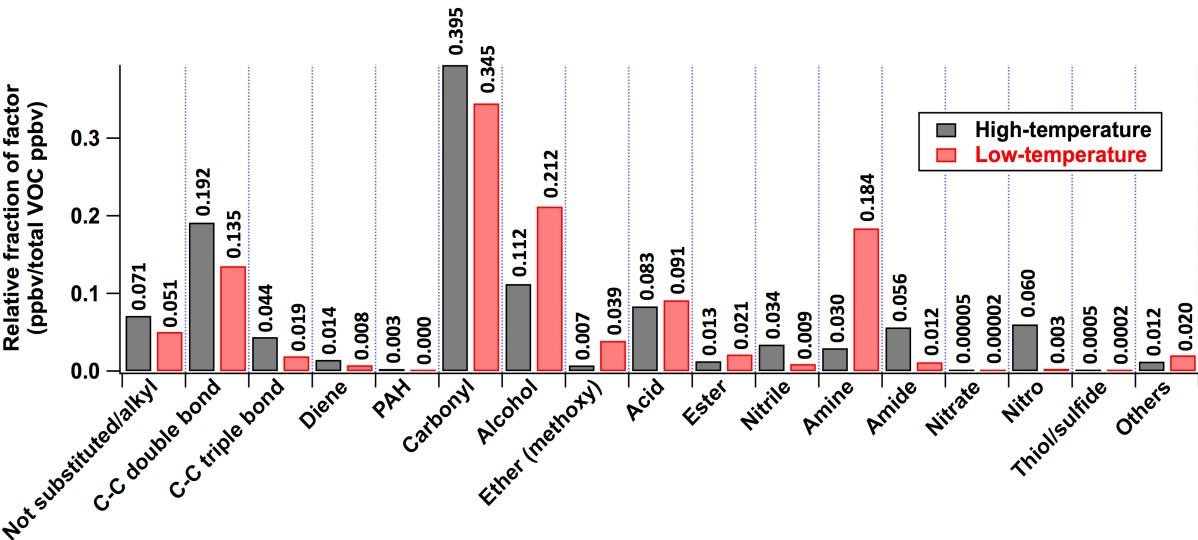

**Figure 8. VOC composition in high- and low-temperature pyrolysis emission profiles (Figure 3) sorted by 17 functional groups. Each group includes various structures and elemental composition. Some VOCs have multiple functional groups. These are counted once in each relevant category. For example, guaiacol is counted in the categories of "Alcohol" and "Ether (methoxy)".**

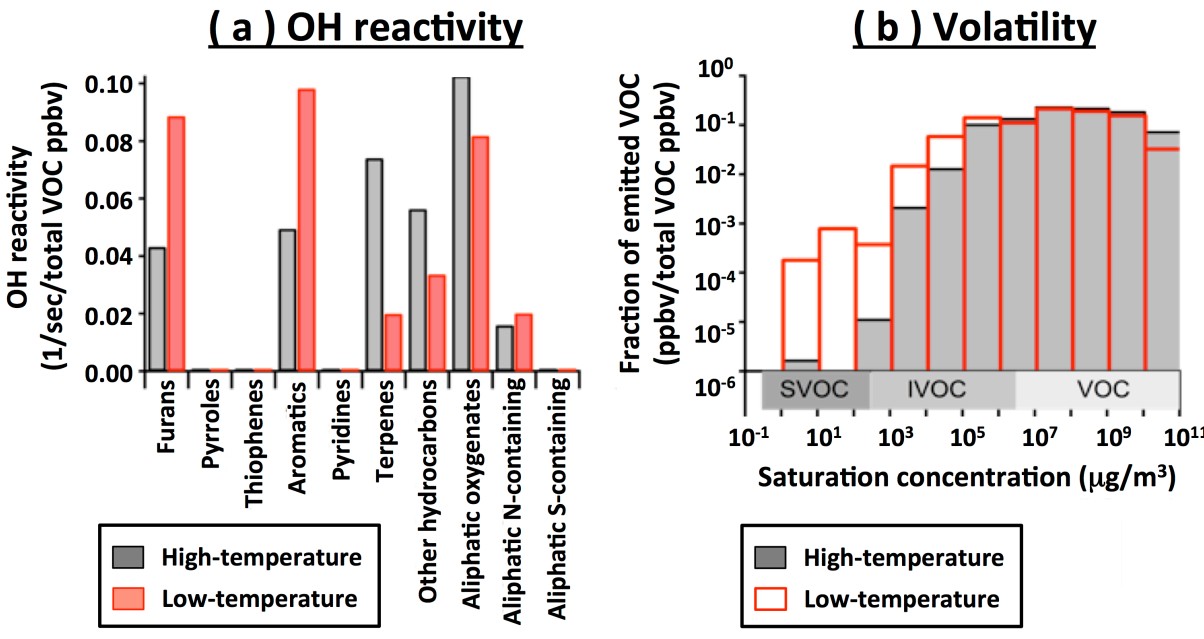

**Figure 9. High- and low-temperature emission profiles compared by (a) OH reactivity and (b) volatility, described by saturation concentration (μg m⁻³).**

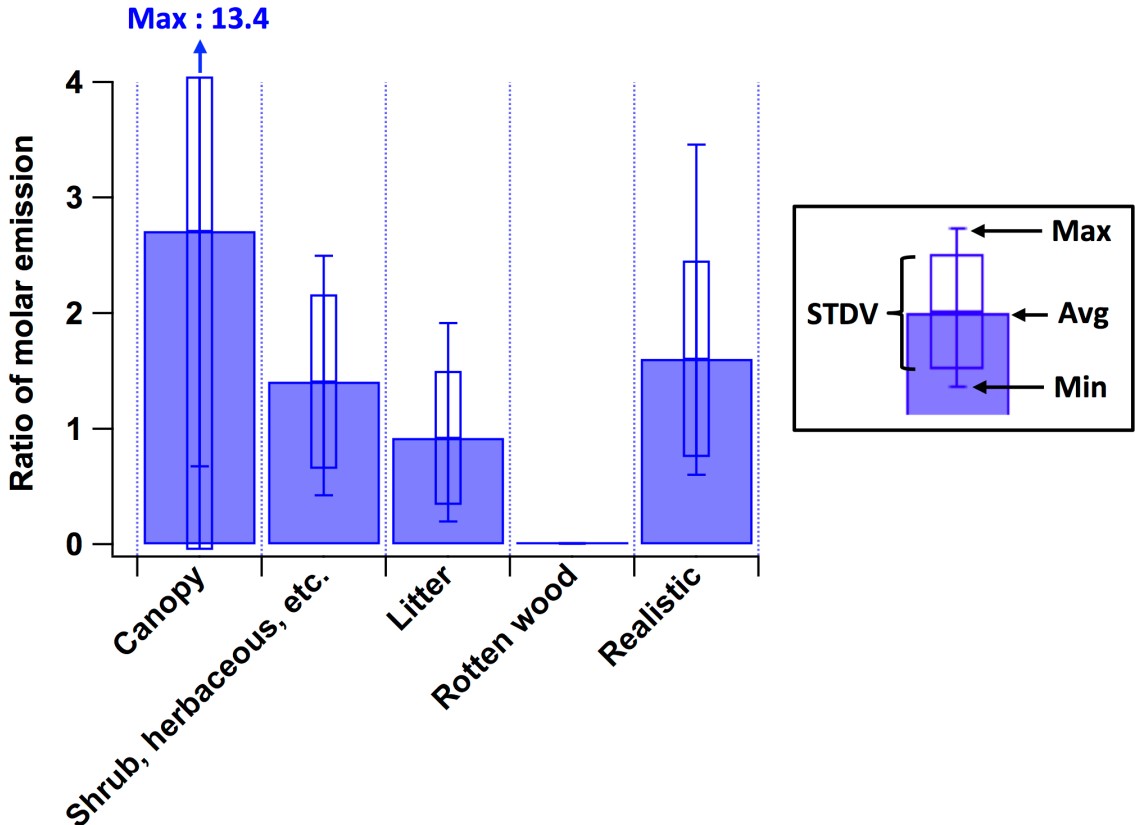

**Figure 10.** Ratios of fire-integrated molar emissions of total VOCs from high- to low-temperature pyrolysis ("$\sum VOC_{High\text{-}T}/\sum VOC_{Low\text{-}T}$") for different type fuel parts, obtained using PMF results of 15 different fuels.

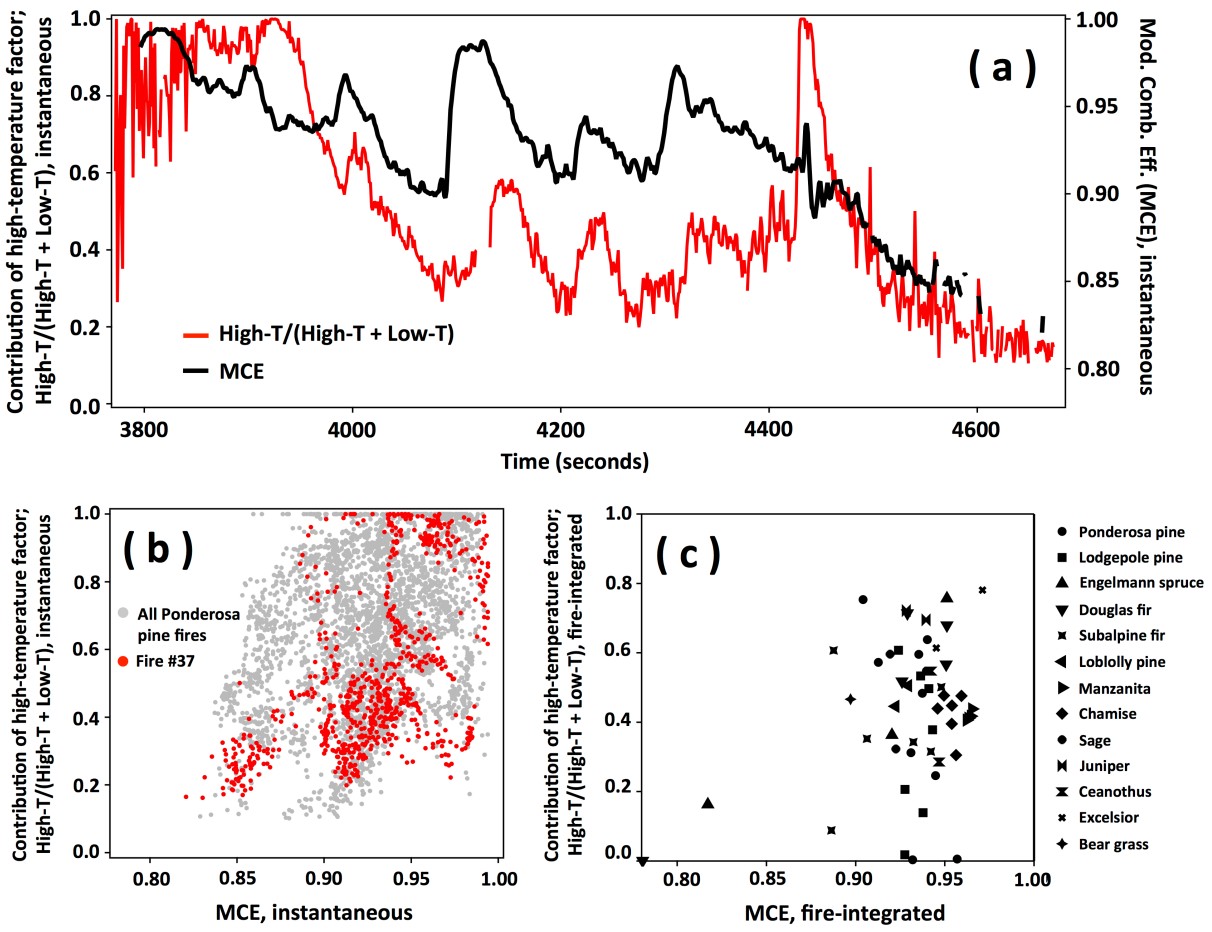

**Figure 11.** The comparison of contribution of high-temperature factor versus modified combustion efficiency (MCE). (a) Time series of Fire #37 (Ponderosa pine realistic mixture). (b) Scatter plot of instantaneous high-temperature contribution versus MCE for all Ponderosa pine fires. (c) Scatter plot of fire-integrated high-temperature contribution versus MCE for all fires. Contribution of high-temperature factor was calculated by $\Sigma VOC_{high\text{-}T}/(\Sigma VOC_{high\text{-}T} + \Sigma VOC_{low\text{-}T})$ instantaneously or on a fire-integrated basis.

## ( a )  VOC emission profile of duff burn

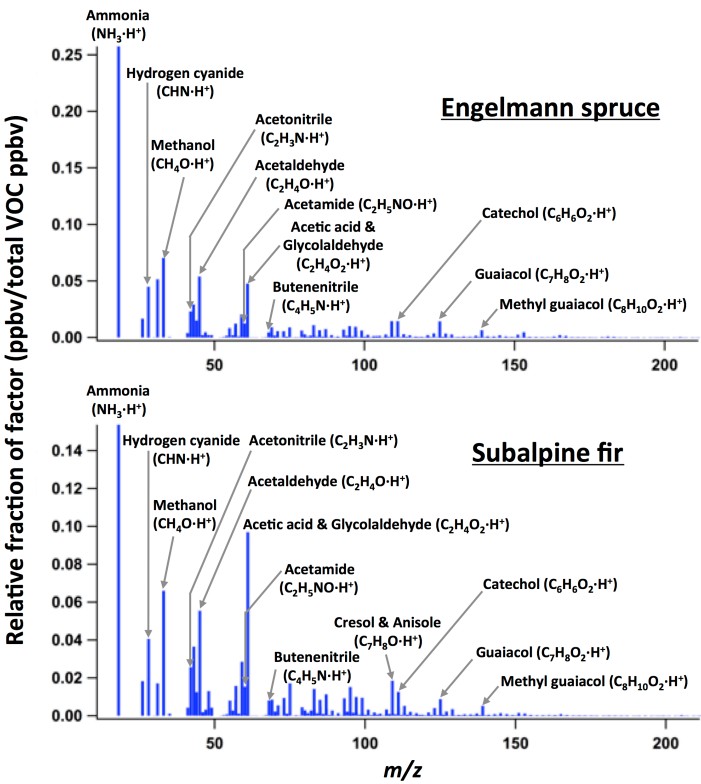

## ( b )  Duff profile vs. Low-temp. profile

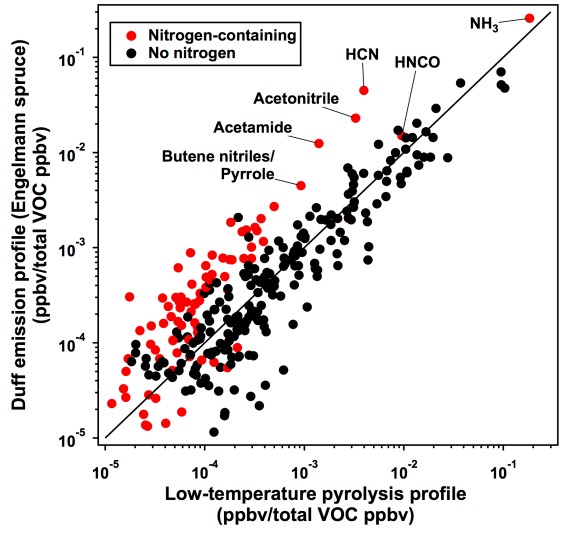

**Figure 12. (a) VOC emission profile of duff burn of Engelmann spruce and Subalpine fir. (b) Scatter plot of duff emission profile (Engelmann spruce) versus average low-temperature pyrolysis profile.**