# Peer review of "organic compound emissions from western US wildfire fuels"

_Atmospheric Chemistry and Physics, 2018_

## Referee Comment (RC1) · Anonymous Referee #1 · 31 Mar 2018

This paper presents results of positive matrix factorization (PMF) analysis on data collected with a Proton Transfer Reaction Mass Spectrometer (PTR-MS) during laboratory burns of various wildfire fuels during the FIREX campaign. The data set, describing VOCs and other volatile components (e.g. NH3), was described in another publication, and here the authors use PMF to show that much of the variability in emission profiles across and within burns can be explained by two factors, which they associated with high- and low-temperature pyrolysis processes. They show that a single pair of factors can explain the variability in most burns nearly as well as fuel/burn-specific factors, with

a few exceptions (e.g. rotten wood, duff). They then included a detailed dive into the two factors, including: the absolute and relative contributions of different compounds, function groups represented in each, and their estimated OH reactivities and volatilities. The paper also emphasizes that modified combustion efficiency (MCE), often used to parameterize combustion types and VOC emission profiles, does not capture the pyrolysis-temperature-driven variability captured here, suggesting that it is not a good proxy for varying emission conditions.

This is really nice paper, which makes excellent use of this interesting data set to put forward a compelling case for the importance of pyrolysis temperature on dictating the mixture of VOCs emitted from biomass combustion. This is a topic of great interest, as wildfire VOC emissions (and their variability across fires, space and time) are important and poorly constrained inputs for atmospheric models. The analysis is thorough, cutting across a multi-dimensional data set in a logical way, and the paper is clearly written and includes highly informative figures. Therefore, this paper is highly suitable for publication in ACP. Below, I list a number of questions and minor concerns that I would like to see addressed before the paper's final publication. Most are clarifications, though a number of points are suggestions for additional steps that the authors may wish to take to enhance the impact and usability of their findings in the broader community interested in biomass burning impacts on the atmosphere.

General comments

Although it was eventually clear, I found the initial attribution of factors to high- and low-temperature pyrolysis processes to be a bit confusing. I think it might be helpful to move Fig. 4b and some more background discussion of combustion processes to the introduction of the paper. There is currently a nice brief introduction (e.g. L60-71), but then the importance of pyrolysis temperature on VOC composition isn't really discussed into well after results are presented (i.e. after discussion of Figs. 1-3). As it is, this feels like a 'grand unveiling' of something that would be better described earlier. For example, line 94 calls high and low-T pyrolysis the 'main processes (sic) of the

VOC emissions from biomass burning' without a reference – if this is so, shouldn't that be made clear earlier, with specific references and summary of what is known about VOC emission as a function of pyrolysis temperature outlined early on? This point is also strongly made on Lines 232-234 without much specific justification.

Some of the description of VOC quantification is a bit unclear. For example – Line 87 says that 90% of instrument signal could be attributed to identified VOCs, but then on lines 159 and 160 it is stated that 'many ion masses cannot be unambiguously related to a single VOC contributor... and cannot be converted to a mixing ratio'. I assume this is because most of the overall signal is due to a relatively small number of compounds, but perhaps this can be clarified a bit – e.g. how many compounds are actually quantified as mixing ratios? Also, can you estimate approximately what fraction of total emitted NMVOC your measurements represent?

The use of the Pearson's correlation coefficient (r) doesn't really seem appropriate since you are actually presenting regression results (slopes) not correlations alone. Also, the Figure 2 caption states that r was calculated based on log-transformed normalized signals, were slopes also calculated this way? This should be done consistently and described clearly. I'm not an expert statistician, but it seems that there may be a better way to compare factors – as it is, you are using difference in r values between 0.91 and 0.83 (for example) to say that factors are quite similar or quite different (Fig. 2d and 2e). I wonder if there are better ways to contrast the spectra than is presented here?

As noted in a few places, one of the major drivers for VOC speciation is to understand SOA formation potential, but the final step to that isn't taken. As stated (L 398) yields are an important missing piece here, other similar papers on biomass combustion has taken that step (Bruns et al. 2016, 2017). The authors could consider adding some kind of scoping analysis to this paper, or certainly extending this in future work with this dataset.

[Figure]

One of the main reasons that MCE appeals as a proxy for combustion conditions is that it is based on easily-measurable quantities (including from satellites). While you may point out its limits here, it would be great if (relatively) simple alternatives were proposed/discussed. For example, are there any especially robust 'marker species' for the two factors across all fuel/burn types? For example, Cresol and guaiacol seem to be distinct features of high and low T factors with substantial contributions (Fig. 2), but perhaps there are others that are more consistent. Could the ratio of these two serve as a proxy for relative factor strength? How can this approach be simplified to be applied by different analytical approaches or other data sets? Does MCE have any explanatory power for relative factor strength across your data set or in some subsets? Are there any other emission ratios (especially for species that are frequently measured, especially by remote sensing) that do?

Specific points:

Lines 239-243 – This point seems like it could use a bit more justification. To my eye, the addition of a third factor in Fig. S2 does seem to make a pretty substantial difference – there is quite a bit of scatter in the fit vs. reconstructed Factor 3 time series plot (bottom right). Is there a way to make this point more convincingly?

Line 337-339 – This is a bit confusing/misleading – pyrolysis will still happen during flaming combustion as there is heat transfer from the flame to surrounding biomass. As you state elsewhere, all of these processes are happening simultaneously in most cases. Therefore, it might be expected that the dominance of different factors might be linked to flaming- versus smoldering-dominated burns or stages of a burn.

Line 232 - should be 'make a higher contribution in the low-T factor', I think?

Line 391-392 – This doesn't seem necessary considering that your estimated OH reactivities for the two factors are basically identical. This is a nice result, that suggests that OH reactivity scales directly with NMVOC (for the compounds you've detected). Seems like something to note.

Section 3.3.3 – It would seem more appropriate to show this distribution on a mass (versus molar) basis, as is typically done for volatility basis set representations. As noted above, yields could be approximated to take the next step, which would be very helpful but not necessary.

Line 445 -450 – I see this is true across fuel/burn types, but what about within a burn? Are there any stages in which transitions in MCE map to transitions in factor-strength? The time-series discussed here shows CO/CO2 for the rotten pine case (Fig. S6), but that doesn't seem like an appropriate example as it is pretty much dominated by the low-T factor (I can't visually average these two...). Having some time series plots with MCE during a burn along with the factor contributions (possibly a separate trace on Fig. 1) would address this better. Is there any correlation between MCE and the relative contribution from the different factors during burns?

Line 558-560- Residuals from PMF fitting are not really discussed elsewhere in the paper. They either should be or the effectiveness of fitting discussed in another way here.

Figure 4b – Would like a more detailed caption. What do color bars correspond to - just temperature ranges? Why does the red bar span a wider temperature range for lignin?

References

Bruns, E. A., El Haddad, I., Slowik, J. G., Kilic, D., Klein, F., Baltensperger, U., and Prévôt, A. S. H. (2016). "Identification of significant precursor gases of secondary organic aerosols from residential wood combustion." Scientific Reports, 6, 27881.

Bruns, E. A., Slowik, J. G., El Haddad, I., Kilic, D., Klein, F., Dommen, J., Temime-Roussel, B., Marchand, N., Baltensperger, U., and Prévôt, A. S. H. (2017). "Characterization of gas-phase organics using proton transfer reaction time-of-flight mass spectrometry: fresh and aged residential wood combustion emissions." Atmos. Chem. Phys., 17(1), 705–720.

---

## Referee Comment (RC2) · Anonymous Referee #2 · 2 Apr 2018

General Comments:

The paper by Sekimoto et al. seeks to understand the "instantaneous" variability in VOC emissions from biomass burning, to develop predictive capability of the emissions. The authors report that a PMF solution consisting of just two emission profile factors can explain on average 85% of the VOC emissions across various fuels representative of the western US. They state that the profiles are remarkably similar across almost all of the fuel types tested. According to the authors, the two factor solution from the PMF model could be attributed to "high-temperature" and "low-temperature"

[Figure]

pyrolysis processes. They suggest that this type of temperature based parameterization of emissions could be widely useful to model VOC emissions from many types of biomass burning in the western US, with exceptions such as burns of duff and rotten wood. Certainly this is a very interesting topic and the idea of being able to predict emissions resulting from complex combustion chemistry using temperature regimes is indeed appealing. As an idea I do find the paper novel. However, there appear to be major shortcomings in the current version of the paper and analyses.

Thus, I was not able to conclude that the evidence presented in the paper is sufficient and convincing to have confidence in the main take home message of the paper that has been aptly summarized in its title, namely: "High- and low-temperature pyrolysis profiles describe volatile organic compound emissions from western US wildfire fuels"

Major reservations and concerns:

1) The present analyses and discussion completely omits the role of oxygen (read air to fuel ratio) during the combustion experiments. I find it illogical that burn conditions such as the oxygen supply when the fuel is being burnt can be completely discounted from playing any role. The major advantage of the modified combustion efficiency (MCE) developed by some of the senior authors in their previous works is that it is able to capture the role of oxygen availability between smoldering and flaming dominated combustion conditions and experimental evidence in the form of the delta CO/delta CO2 ratios helps us reconcile it with the real world process of the overall oxidation of the carbon in the fuel (e.g. incomplete oxidation implies more CO and reduced carbon compounds will be emitted). Though the authors show in Fig 10, that the ratio of high temperature to low temperature emissions shows poor dependency on the MCE, it would have been more convincing to show this using the online real time data of the burns. For example, the authors could add the time series data of CO and CO2 for the same fire as additional panels to Figure 1. This would help distinguish whether the new terminology of high and low temperature pyrolysis are simply a variant of the older terminology of flaming and smoldering fire stages. To put it another way, will the high/low

temperature burn emissions of the fuel under low and high oxygen conditions show similar profiles and behavior? Knowing the important role played by oxygen in flame chemistry, how can such an important aspect be ignored from the parameterization? If the authors think otherwise, then some experimental data in support of their contention is certainly warranted else one cannot accept the parameterization proposed by the authors as generically as they suggest.

2) The justification for the two factor PMF solution is not at all clear. Figure 3 should be modified to show the average VOC emission profiles of both the high temperature and low temperature factor for a PMF solution of 2 factors, 3 factors and 4 factors respectively, in the same panel (as three different colored lines for each m/z). In addition, the same should also be shown for the third and fourth factors. The issue with the correlation plots given in the supplement is that they mask more than they reveal whereas in a VOC emission profile of the type shown in Fig 3, specific information on which compounds migrate from one factor to a new factor can be clearly made out and is more helpful for assessing whether the proposed solution is justified.

3) The paper makes generic assertions about low and high temperature emissions, but in none of the experiments was the temperature data provided/shown or even mentioned. I presume the authors have such data /or can provide it. While I understand different parts of the fuel may be subject to different stages of combustion (flaming, combustion, distillation) at the same time and the emissions are a net resultant of these, some temperature data from the burn experiments where a large temperature range was observed from start to finish is certainly in order.

4) The authors should clarify/mention the practical steps that need to be followed to employ the high and low temperature pyrolysis profiles for modeling VOC emissions from "many types of biomass burning" as asserted in the abstract and conclusion.

Some specific details:

Line 136-138: Citing Selimovic et al. 2017 for important and relevant details is not

sufficient in my opinion. For this paper to be considered as an independent paper, at least a summary of the essential info needs to be provided in the main text. For example, how many fires for each fuel type were sampled in the present work? how similar / dissimilar were the emission profiles when the same fuel type was burnt? Was the pyrolysis temperature or temperature profile during the burns measured? What about the variability induced by other burn conditions?

Further discussion points:

The authors need to elaborate and clarify regarding the limitations of their methodology. In particular, is the interpretation of the data resting just on the statistical treatment, correlation with known products emitted at certain stages of combustion and a hypothesized area/volume ratio? If so, how can these findings be extrapolated to scenarios in the real world which differ from the experimental conditions of the present work? How would other known factors which are missing from their interpretation and analyses such as oxygen (air to fuel ratio variability) and/or moisture content complicate or change the predicted emission behavior? In particular, if emissions resulting from burns of duff and rotten wood are not explained well by their present predictive algorithm, could variability of conditions significantly alter VOC emission profiles of the other currently "well explained" fuels.

MINOR/TECHNICAL COMMENTS

The language and presentation is some parts of the MS can be improved with more careful reading. Two examples are mentioned below: Abstract: Line 26: Precursors to the formation.... " the formation of.." is redundant.. Line 27: Measurements collected.... Measurements performed?

---

## Author Comment (AC1) · 28 May 2018

Journal: Atmospheric Chemistry and Physics
Title: High- and low-temperature pyrolysis profiles describe volatile organic compound emissions from western US wildfire fuels
Author(s): Kanako Sekimoto et al.
MS No.: acp-2018-52
MS Type: Research article

**Response to Reviewers:**

We thank both reviewers for their positive comments and helpful feedback. The manuscript has been revised accordingly. Our responses are written in blue text.

=====
**Reviewer 1**

This paper presents results of positive matrix factorization (PMF) analysis on data collected with a Proton Transfer Reaction Mass Spectrometer (PTR-MS) during laboratory burns of various wildfire fuels during the FIREX campaign. The data set, describing VOCs and other volatile components (e.g. NH3), was described in another publication, and here the authors use PMF to show that much of the variability in emission profiles across and within burns can be explained by two factors, which they associated with high- and low-temperature pyrolysis processes. They show that a single pair of factors can explain the variability in most burns nearly as well as fuel/burn-specific factors, with a few exceptions (e.g. rotten wood, duff). They then included a detailed dive into the two factors, including: the absolute and relative contributions of different compounds, function groups represented in each, and their estimated OH reactivities and volatilities. The paper also emphasizes that modified combustion efficiency (MCE), often used to parameterize combustion types and VOC emission profiles, does not capture the pyrolysis-temperature-driven variability captured here, suggesting that it is not a good proxy for varying emission conditions. This is really nice paper, which makes excellent use of this interesting data set to put forward a compelling case for the importance of pyrolysis temperature on dictating the mixture of VOCs emitted from biomass combustion. This is a topic of great interest, as wildfire VOC emissions (and their variability across fires, space and time) are important and poorly constrained inputs for atmospheric models. The analysis is thorough, cutting across a multi-dimensional data set in a logical way, and the paper is clearly written and includes highly informative figures. Therefore, this paper is highly suitable for publication in ACP. Below, I list a number of questions and minor concerns that I would like to see addressed before the paper's final publication. Most are clarifications, though a number of points are suggestions for additional steps that the authors may wish to take to enhance the impact and usability of their findings in the broader community interested in biomass burning impacts on the atmosphere.

We thank the reviewer for their positive comments. Below we have responded to specific comments.
* * *
General comments
Although it was eventually clear, I found the initial attribution of factors to high- and low-temperature pyrolysis processes to be a bit confusing. I think it might be helpful to move Fig. 4b and some more background discussion of combustion processes to the introduction of the paper. There is currently a nice brief introduction (e.g. L60-71), but then the importance of pyrolysis temperature on VOC composition isn't really discussed into well after results are presented (i.e. after discussion of Figs. 1-3). As it is, this feels like a 'grand unveiling' of something that would be better described earlier. For example, line 94 calls high and low-T pyrolysis the 'main processes (sic) of the VOC emissions from biomass burning' without a reference – if this is so, shouldn't that be made clear earlier, with specific references and summary of what is known about VOC emission as a function of pyrolysis temperature outlined early on? This point is also strongly made on Lines 232-234 without much specific justification.

Edits were made in the abstract and introduction to better introduce the relationship between VOC composition and temperature. At line 40, the abstract is revised to read,

"**The compositional differences between the two VOC profiles appear to be related to differences in pyrolysis processes of fuel biopolymers at high and low temperatures. These pyrolysis processes are thought to be the main source of VOC emissions**."

At line 78, appropriate references are added and the introduction is revised to read,

"The main source of VOC emissions is pyrolysis of the polymers that form biomass such as cellulose, hemicellulose, and lignin. **The temperature of the reaction and the physical characteristics of the biopolymer control which pyrolysis mechanism (e.g. depolymerization, fragmentation, or aromatization) is the main source of emitted VOCs (Yokelson et al., 1996; Yokelson et al., 1997; Collard and Blin, 2014; Liu et al., 2016).** In a given fire, the processes (i)-(iv) occur simultaneously, but the relative importance of each process and temperature can change with time, which relates to the variability in integrated VOC emissions between different fires."

At line 119, we have revised the introduction to read:

"We show that much of the observed variability in VOCs can be explained by only two factors, **and that these two factors are qualitatively related to the temperature of the pyrolysis processes, which are the main sources of the VOC emissions from biomass burning. Based on this result, the two factors are named as** a high-temperature pyrolysis factor and a low-temperature pyrolysis factor."

As Figure 4b (temperature diagram from the literature) includes information that is too specific for the Introduction, we decided to leave it in Results and Discussions as is.
* * *
Some of the description of VOC quantification is a bit unclear. For example – Line 87 says that 90% of instrument signal could be attributed to identified VOCs, but then on lines 159 and 160 it is stated that 'many ion masses cannot be unambiguously related to a single VOC contributor. . . and cannot be converted to a mixing ratio'. I assume this is because most of the overall signal is due to a relatively small number of compounds, but perhaps this can be clarified a bit – e.g. how many compounds are actually quantified as mixing ratios? Also, can you estimate approximately what fraction of total emitted NMVOC your measurements represent?

At line 205, we have added an example of a detected ion that cannot be unambiguously related to a single VOC. We have also added information about how many compounds were quantified, and what fraction of total VOC emissions this might represent. The insertion is:

"**For example, $C_7H_{13}^+$ ($m/z$ 97.101) is a fragmentary product ion of at least five different VOCs, whose relative contributions are different between fires. However, variability in these ion signals still contains information useful for PMF. To interpret the PMF results, we did convert to mixing ratio where possible (Section 2.4). 528 compounds were quantified, of which 156 are identified VOCs. The PTR-ToF-MS measures 50-80% of total emitted non-methane VOC mass, with uncertainty in this value due to semivolatile compounds (Hatch et al., 2017)**."
* * *
The use of the Pearson's correlation coefficient (r) doesn't really seem appropriate since you are actually presenting regression results (slopes) not correlations alone. Also, the Figure 2 caption states that r was calculated based on log-transformed normalized signals, were slopes also calculated this way? This should be done consistently and described clearly. I'm not an expert statistician, but it seems that there may be a better way to compare factors – as it is, you are using difference in r values be-tween 0.91 and 0.83 (for example) to say that factors are quite similar or quite different (Fig. 2d and 2e). I wonder if there are better ways to contrast the spectra than is presented here?

Both the slope and correlation coefficient were calculated using log-transformed normalized signals. This is now stated clearly in the caption of Figure 2. We looked into other ways of determining correlation coefficients

between the spectra. Pearson's correlation coefficient is the most appropriate correlation coefficient to use here. The use of $R^2$ does make a better distinction between the spectra that are better correlated (Figure 2 panels a-d) and not as well correlated (Figure 2 panel e), so we have updated Figure 2, Figure S4, Table S1, and the text to use $R^2$. Our assessment of correlation is not based on R alone, but also on slope (closer to 1 for the spectra that correlate), and visual inspection of Figure 2 and the other pairs of spectra. Figure 2e shows clearly worse correlation between high-and low-temperature profiles for a single fuel, than between high-temperature (or low-temperature) profiles of different fuels.
* * *
As noted in a few places, one of the major drivers for VOC speciation is to understand SOA formation potential, but the final step to that isn't taken. As stated (L 398) yields are an important missing piece here, other similar papers on biomass combustion has taken that step (Bruns et al. 2016, 2017). The authors could consider adding some kind of scoping analysis to this paper, or certainly extending this in future work with this dataset.

SOA yields from the emissions described here were studied during the FIREX campaign. The analysis of this experiment is ongoing and we have now noted that in this section. A reference to Bruns et al. (2016) has also been added. Section 3.3.3 "Volatility" now reads (line 492),

"**Oxygenated aromatics have been shown to be important biomass burning SOA precursors (Bruns et al., 2016), and while** the SOA yields of many other compounds are unknown, the lower volatility and higher oxygen content of the low-temperature profile suggests a potentially more efficient SOA formation. **SOA formation was also studied during the FIREX 2016 campaign, by oxidizing emissions in a chamber, and will be presented separately (Lim et al, in prep, 2018)**."
* * *
One of the main reasons that MCE appeals as a proxy for combustion conditions is that it is based on easily-measurable quantities (including from satellites). While you may point out its limits here, it would be great if (relatively) simple alternatives were proposed/discussed. For example, are there any especially robust 'marker species' for the two factors across all fuel/burn types? For example, Cresol and guaiacol seem to be distinct features of high and low T factors with substantial contributions (Fig. 2), but perhaps there are others that are more consistent. Could the ratio of these two serve as a proxy for relative factor strength? How can this approach be simplified to be applied by different analytical approaches or other data sets? Does MCE have any explanatory power for relative factor strength across your data set or in some subsets? Are there any other emission ratios (especially for species that are frequently measured, especially by remote sensing) that do?

We considered several pairs of VOCs that could be used as a proxy for the high vs. low temperature factor strength. Ethyne and furan are a robust pair, although they are not ideal because few instruments other than PTR-MS and GC methods can measure them. We have also added some more discussion of MCE, and an additional figure that shows MCE correlates poorly with factor strength.

At line 545, Section 3.5 has been edited to read and added a new figure (Figure 11),

"However, MCE does not parameterize the relative amounts of high- and low-temperature pyrolysis products very well, **either instantaneously or on a fire-integrated basis (Figure 11)**."

[Figure]

Figure 11. The comparison of contribution of high-temperature factor versus modified combustion efficiency (MCE). (a) Time series of Fire #37 (Ponderosa pine realistic mixture). (b) Scatter plot of instantaneous high-temperature contribution versus MCE for all Ponderosa pine fires. (c) Scatter plot of fire-integrated high-temperature contribution versus MCE for all fires. Contribution of high-temperature factor was calculated by $\int VOC_{high-T}/(\int VOC_{high-T} + \int VOC_{low-T})$ instantaneously or on a fire-integrated basis.

MCE data in Figure 10 has been removed, and now this figure has only the ratio of fire-integrated VOC molar emissions (see below).

[Figure]

Figure 10. Ratios of fire-integrated molar emissions of total VOCs from high- to low-temperature pyrolysis ("$\Sigma VOC_{High-T}/\Sigma VOC_{Low-T}$") for different type fuel parts, obtained using PMF results of 15 different fuels.

At line 571 in Section 3.5, additional discussion of ethyne and furan as markers is inserted:

"**The relative contributions from the high- and low-temperature processes could be estimated from ratios of distinct marker species that are consistently enhanced in the high and low-temperature profiles. Several such pairs were considered and the ratio of ethyne ($C_2H_2$) to furan ($C_4H_4O$) can reasonably predict the ratio of high- to low-temperature emissions as given in Eq. 1:**

$$\frac{total\,VOC\,,high\,temperature\,(ppbv)}{total\,VOC\,,low\,temperature\,(ppbv)} = \frac{ethyne\,(ppbv)\,/\,0.0393}{furan\,(ppbv)\,/\,0.0159} \qquad \textbf{(1)}$$

**The derivation and how the ethyne/furan ratio correlates with the high-/low-temperature emission ratio are given in the Supporting Information (S2 and Figure S9). However, this pair is not ideal because measurements of these two species are not frequently available and furan has high reactivity to both $O_3$ and $NO_3$ radicals. Future work should assess non-PTR measurements in order to find appropriate external markers.**"

[Figure]

Figure S9. The comparison of contribution of high-temperature factor versus ethyne/furan ratio. (a) Time series of Fire #37 (Ponderosa pine realistic mixture). (b) Scatter plot of instantaneous high-temperature contribution versus ethyne/furan ratio for all Ponderosa pine fires. (c) Scatter plot of fire-integrated high-temperature contribution versus ethyne/furan ratio for all fires. Contribution of high-temperature factor was calculated by ∫VOC_high-T/(∫VOC_high-T + ∫VOC_low-T) instantaneously or on a fire-integrated basis.

The relevant section in the Supporting Information reads:

**"S2. Relationship of ethyne:furan ratio to high:low temperature ratio**

Trace gases can be used to estimate the emissions from the high/low temperature factors. Here we propose ethyne ($C_2H_2$) and furan ($C_4H_4O$) as tracers. Normalized fractions of the high/low temperature factors are 72%/28% for ethyne and 33%/67% for furan. These two compounds have large emissions and low standard deviations in the average emission profiles of 15 different fuels (0.0393 ± 23% ppbv for ethyne in the high-temperature factor and 0.0159 ± 19% ppbv for furan in the low-temperature factor). This reduces to a ratio of approximately:

$$\frac{total\ VOC, high\ temperature\ (ppbv)}{total\ VOC, low\ temperature\ (ppbv)} = \frac{ethyne\ (ppbv)\ /\ 0.0393}{furan\ (ppbv)\ /\ 0.0159} \quad \text{(S2)}$$

Average relative error (%) of the ethyne/furan ratio to the total VOC_High-T/total VOC_Low-T is 50%, except for rotten wood."
* * *
Specific points:
Lines 239-243 – This point seems like it could use a bit more justification. To my eye, the addition of a third factor in Fig. S2 does seem to make a pretty substantial difference – there is quite a bit of scatter in the fit vs. reconstructed Factor 3 time series plot (bottom right). Is there a way to make this point more convincingly?

According to comments from both reviewers, we have significantly modified Figures S2 and S3 to more clearly show that a two-factor solution is reasonable, and that additional factors do not add useful information.

Figure S2 now shows the mass spectra from the 2-, 3-, and 4- factor solutions. The two factors that account for most of the variability (Factor 1 and Factor 2) do not change substantially as the number of factors increases. The additional factors in the 3- and 4- factor solutions have similar mass spectra to Factor 1 and Factor 2.

Figure S3 now shows the time series of a representative fire of Ponderosa pine. The measured signal is compared to the PMF solution with two, three, and four factors. For each solution (2, 3, and 4-factors), the time series of the individual factors are shown as subplots. In each subplot, the linear best fit of the high- and low-temperature factors is shown. It can now be clearly seen that there is low residual with the 2-factors solution; additional factors do not substantially change the residual; and that each factor from the 3- and 4-factor solutions can be described by a linear combination of the high- and low- temperature factors.

The two new figures are as shown:

[Figure]

Figure S2. Comparison of mass spectra from 2-, 3-, and 4-factor PMF solutions. The two factors that account for most of the variability (Factor 1 and Factor 2) do not change substantially as the number of factors increases. The additional factors in the 3- and 4-factor solutions have similar mass spectra to Factor 1 and Factor 2.

**A. PMF factor fit to measured signal**

**B. High- and low-temperature factors linear fit to other possible solutions**

*2-factor solution*
High-temp. factor
Low-temp. factor
Measured signal

*3-factor solution*
Factor 1
Factor 2
Factor 3
Measured signal

F1=0.9264(High-T)
*r* = 0.9988

F2=0.0507(High-T)
+ 0.7063(Low-T)
*r* = 0.9200

F3=0.0224(High-T)
+ 0.3466(Low-T)
*r* = 0.8432

*4-factor solution*
Factor 1
Factor 2
Factor 3
Factor 4
Measured signal

F1=0.7966(High-T)
*r* = 0.9862

F2=0.0772(High-T)
+ 0.5565(Low-T)
*r* = 0.8786

F3=0.4280(High-T)
*r* = 0.9862

F4=0.1904(High-T)
+ 0.1241(Low-T)
*r* = 0.7137

Figure S3. Comparison of time series from 2-, 3-, and 4-factor solutions. The time-series shown is the total instrument signal from a representative fire of Ponderosa Pine (a part of Fire #37). The individual factors are from the PMF analysis of the extended time series (in which all fires of Ponderosa Pine were concatenated). The left-side plots show the stacked contributions of the factors, compared to the measured signal. The small plots on the right show the time series of the individual factors (solid lines). The high- and low-temperature factors were fit to each factor in the 3- and 4-factor solutions. The best-fit was done using the extended time series. These best-fits are shown as the shaded areas in the right-side plots, and the best-fit equation and correlation coefficient are also provided.
* * *
Line 337-339 – This is a bit confusing/misleading – pyrolysis will still happen during flaming combustion as there is heat transfer from the flame to surrounding biomass. As you state elsewhere, all of these processes are happening simultaneously in most cases. Therefore, it might be expected that the dominance of different factors might be linked to flaming- versus smoldering-dominated burns or stages of a burn.

At the end of Section 3.2 (line 424), the description has been revised to read,
"The present analysis predominantly focuses on VOCs. The VOC emissions from biomass burning are dominated by pyrolysis reactions of biopolymers. However, **not all species are emitted from pyrolysis reactions**. For example, flaming combustion releases $CO_2$, $NO_x$, HONO, and black carbon, etc. This is a separate process and cannot be expected to be captured by our VOC framework. **In section 3.5 we show that MCE, which delineates flaming versus smoldering combustion, is a poorer descriptor of VOC variability than the high versus low-temperature pyrolysis framework**."

We also added the following figure as Figure S6 in the Supporting Information, in order to show that the emissions of compounds originating from flaming combustion (i.e., $CO_2$ and $NO_x$) do not correlate with the high- and low-temperature pyrolysis factors.

[Figure]

[Figure]

Figure S6. Linear fits of (a) $CO_2$, (b) $NO_x$, and (c) CO emissions (in ppmv) by the high- and low-temperature pyrolysis time series (in ncps) for Fire #37 (Ponderosa pine realistic mixture). Each plot shows the stacked contributions of the high- and low-temperature factors (shaded area), compared to the measured mixing ratios (solid line). The best-fit equation and correlation coefficient are also provided.
* * *
Line 232 - should be 'make a higher contribution in the low-T factor', I think?

Line 373 has been clarified to read,
   "**Emissions** of these compounds have a **larger** contribution from the low-temperature factor ($F_{low-T}$ = 60-100%)."
* * *
Line 391-392 – This doesn't seem necessary considering that your estimated OH reactivities for the two factors are basically identical. This is a nice result, that suggests that OH reactivity scales directly with NMVOC (for the compounds you've detected). Seems like something to note.

At line 484, this sentence has been revised to read,
   "Since the total VOC emissions in real-world fires come from a mixture of the high- and low- temperature pyrolysis factors, the total OH reactivity **of fresh emissions should scale directly with VOC concentration**."
* * *
Section 3.3.3 – It would seem more appropriate to show this distribution on a mass (versus molar) basis, as is typically done for volatility basis set representations. As noted above, yields could be approximated to take the next step, which would be very helpful but not necessary.

Figure 9b has been updated to show volatility distribution on a mass basis, as follows:

[Figure]
* * *
Line 445 -450 – I see this is true across fuel/burn types, but what about within a burn? Are there any stages in which transitions in MCE map to transitions in factor-strength? The time-series discussed here shows CO/CO2 for the rotten pine case (Fig. S6), but that doesn't seem like an appropriate example as it is pretty much

dominated by the low-T factor (I can't visually average these two. . .). Having some time series plots with MCE during a burn along with the factor contributions (possibly a separate trace on Fig. 1) would address this better. Is there any correlation between MCE and the relative contribution from the different factors during burns?

MCE does not parameterize the relative amounts of high- and low-temperature pyrolysis products very well, either instantaneously or on a fire-integrated basis. This is because emissions of $CO_2$ (and $NO_x$) from flaming combustion do not correlate with a linear combination of the high- and low-temperature pyrolysis processes, as shown in "Figure S6" newly added (please see above).

According to comments from both reviewers, we have also added some more discussion of MCE, and an additional figure that shows MCE correlates poorly with factor strength. Section 3.5 has been edited to read (line 545) and added a new figure (Figure 11),

"However, MCE does not parameterize the relative amounts of high- and low-temperature pyrolysis products very well, **either instantaneously or on a fire-integrated basis (Figure 11)**."

The new Figure 11 is shown in our response to the general comments from this reviewer.

According to the reviewer's comment, we added the time series of $CO_2$, CO, $NO_x$, and MCE in Figure 1; also, the corresponding sentences in Section 3.1. have been revised to read (line 285),

"**On the contrary, emissions of compounds mainly from flaming or non-pyrolysis smoldering processes, such as CO, $CO_2$, and $NO_x$ (Figure 1c), do not correlate well with the individual PMF factors (more detailed discussion is given in Section 3.5). This indicates that the two PMF factors do not correspond to the flaming and smoldering combustion processes that are described by MCE and often referenced in biomass burning literature. The main source of VOC emissions is pyrolysis of fuel biopolymers, and *not* the flaming and/or other combustion processes.**"

The revised Figure 1 is as follows:

[Figure]

Figure 1. Results for an example burn of Ponderosa pine realistic mixture (Fire #37). (a) Time series of ion signals of 574 ion peaks, naphthalene ($C_{10}H_8 \cdot H^+$, *m/z* 129.070), and syringol ($C_8H_{10}O_3 \cdot H^+$, *m/z* 155.070). (b) PMF results of 2-factor solution. The grey and pink colors are stacked, not overlapped. (c) Time series of mixing ratios of $CO_2$, CO, and $NO_x$ measured by open-path Fourier transform infrared (OP-FTIR) optical spectroscopy and the modified combustion efficiency (MCE) (Selimovic et al., 2018). The MCE trace is colored by the key and scale on the right.
* * *
Line 558-560- Residuals from PMF fitting are not really discussed elsewhere in the paper. They either should be or the effectiveness of fitting discussed in another way here.

Residuals from PMF are discussed in Section 3.1. We have clarified this through some minor edits in Section 3.1 (line 279):

"These two PMF factors (Figure 1b) describe the total VOC emissions remarkably well for most fuels: **residuals (the differences between the measured ion signals and the calculated ion signals based on the PMF fits)** are less than 15% on average, **except for Douglas fir, Engelmann spruce, and Subalpine fir for which the residual average is 20-25%. The residuals for individual fuels are summarized in** Table 1c."

The residuals from Duff were higher. This was mentioned at the end of Section 3.1 and discussed in detail in Section 3.6.2. The quantitative measure of effectiveness of fitting, "$Q$" is expressed by summation of squared scaled residuals for each experimental data point and is discussed in detail in S1.2.
* * *
Figure 4b – Would like a more detailed caption. What do color bars correspond to - just temperature ranges? Why does the red bar span a wider temperature range for lignin?

The color bars correspond to temperature ranges reported in the literature (Collard and Blin, 2014). According to the reviewer's comment, a more detailed caption was added to Figure 4, and now reads,
  "Figure 4. (a) Normalized fraction of factors for selected biomass pyrolysis products, obtained using PMF results of 15 different fuels. (b) Diagram of the relationship between pyrolysis temperature and products for hemicellulose, cellulose, and lignin, **as reported in the literature (Collard and Blin, 2014). Individual color bars show the temperature range to form specific products described by chemical structures.**"

=====
**Reviewer 2**
General Comments:
The paper by Sekimoto et al. seeks to understand the "instantaneous" variability in VOC emissions from biomass burning, to develop predictive capability of the emissions. The authors report that a PMF solution consisting of just two emission profile factors can explain on average 85% of the VOC emissions across various fuels representative of the western US. They state that the profiles are remarkably similar across almost all of the fuel types tested. According to the authors, the two factor solution from the PMF model could be attributed to "high-temperature" and "low-temperature" pyrolysis processes. They suggest that this type of temperature based parameterization of emissions could be widely useful to model VOC emissions from many types of biomass burning in the western US, with exceptions such as burns of duff and rotten wood. Certainly this is a very interesting topic and the idea of being able to predict emissions resulting from complex combustion chemistry using temperature regimes is indeed appealing. As an idea I do find the paper novel. However, there appear to be major shortcomings in the current version of the paper and analyses. Thus, I was not able to conclude that the evidence presented in the paper is sufficient and convincing to have confidence in the main take home message of the paper that has been aptly summarized in its title, namely: "High- and low-temperature pyrolysis profiles describe volatile organic compound emissions from western US wildfire fuels"

We thank the reviewer for their positive comments, and hope we have sufficiently addressed their reservations in the following response. Overall, we stress that we do not claim to have developed a quantitative temperature-based parameterization that can predict VOC emissions in any scenario. To clarify this, we have edited sections of the abstract, introduction, and other parts of the paper. The edits are intended to more clearly convey the major points of this paper:
  (1) We have determined that there are two distinct emission profiles. These can describe most VOC compositional variability.
  (2) We have evidence that strongly suggests the profiles have a physically realistic explanation (temperature of the pyrolysis process).
  (3) Because the two profiles are likely derived from processes that occur in a wide variety of fires, this conceptual view of VOC emissions could be useful beyond this laboratory study.

Specifically, we have added discussion and a figure (now Figure 11) showing that MCE has limitations to describe VOC compositional variability, supplemental figures (Figures S2 and S3) that better support the choice of a two-factor PMF solution, and edits so as not to suggest that we have a fully-quantitative temperature-based parameterization.

Edits to the abstract and introduction are listed here. Edits in other sections of the paper are listed below the reviewer specific concerns.

At line 40, the abstract has been edited to read,
"**The compositional differences between the two VOC profiles appear to be related to differences in pyrolysis processes of fuel biopolymers at high and low temperatures. These pyrolysis processes are thought to be the main source of VOC emissions**."

The last sentence of the abstract has been moved to the conclusions (line 725):
"**With this further work, the VOC profiles could be widely useful to model VOC emissions from many types of biomass burning in the western US, with additions to the framework being needed for fires that burn a lot of duff**."

At line 106, the last paragraph of the introduction now states,
"The aims of this work are to understand the variation in gas-phase emissions both over the course of a fire and on a fire-integrated basis. **Ultimately, this improved understanding of emissions variability could be used to** simplify predictions of the emission of secondary organic aerosol (SOA) and ozone precursors."
* * *
Major reservations and concerns:
1) The present analyses and discussion completely omits the role of oxygen (read air to fuel ratio) during the combustion experiments. I find it illogical that burn conditions such as the oxygen supply when the fuel is being burnt can be completely discounted from playing any role. The major advantage of the modified combustion efficiency (MCE) developed by some of the senior authors in their previous works is that it is able to capture the role of oxygen availability between smoldering and flaming dominated combustion conditions and experimental evidence in the form of the delta CO/delta CO2 ratios helps us reconcile it with the real world process of the overall oxidation of the carbon in the fuel (e.g. incomplete oxidation implies more CO and reduced carbon compounds will be emitted). Though the authors show in Fig 10, that the ratio of high temperature to low temperature emissions shows poor dependency on the MCE, it would have been more convincing to show this using the online real time data of the burns. For example, the authors could add the time series data of CO and CO2 for the same fire as additional panels to Figure 1. This would help distinguish whether the new terminology of high and low temperature pyrolysis are simply a variant of the older terminology of flaming and smoldering fire stages. To put it another way, will the high/low temperature burn emissions of the fuel under low and high oxygen conditions show similar profiles and behavior? Knowing the important role played by oxygen in flame chemistry, how can such an important aspect be ignored from the parameterization? If the authors think otherwise, then some experimental data in support of their contention is certainly warranted else one cannot accept the parameterization proposed by the authors as generically as they suggest.

First we clarify a misunderstanding. There is no evidence of oxygen depletion at low MCE or high CO/CO2 (see Akagi et al 2011). In open biomass burning the air can rush in and supply plenty of O2. Low MCE indicates that the amount of pyrolysis or glowing products are high compared to flaming products (Yokelson et al., 1996), which can happen for a variety of reasons, but often related to fuel geometry. PMF examines the pyrolysis products in more detail than is possible with MCE alone. Flaming chemistry was not studied in this paper.
    From this comment, we understand that the reviewer would like to know if the instantaneous MCE has any correlation to the relative intensity of the high- and low-temperature profiles **(1)**. The reviewer also is curious how specific burn conditions quantitatively affect the ratio between the two profiles, and whether the high- / low-temperature description accounts for variation in burn conditions **(2)**. Finally, the reviewer is understandably hesitant to move away from MCE as a descriptor of emissions variability because the connection between the experimental measure of MCE (delta CO/delta $CO_2$) and the fire characteristic it describes (flaming vs. smoldering) is grounded in a well-understood physical process (oxidation extent of the fuel) **(3)**.

**(1)** We first show more clearly that MCE is a poor predictor of the high- / low-temperature factor balance. Both reviewers requested this.

First, we have edited Figure 1 as the reviewer suggested. There is now a third panel that shows the time series of $CO$, $CO_2$, $NO_x$, and MCE for an example fire. We changed the example fire from Fire #02 (ponderosa pine, realistic mixture) to Fire #37 (also ponderosa pine, realistic mixture) because there was incomplete CO, $CO_2$, and $NO_x$ data for Fire #02.

[Figure]

Figure 1. Results for an example burn of Ponderosa pine realistic mixture (Fire #37). (a) Time series of ion signals of 574 ion peaks, naphthalene ($C_{10}H_8 \cdot H^+$, *m/z* 129.070), and syringol ($C_8H_{10}O_3 \cdot H^+$, *m/z* 155.070). (b) PMF results of 2-factor solution. The grey and pink colors are stacked, not overlapped. (c) Time series of mixing ratios of $CO_2$, CO, and $NO_x$ measured by open-path Fourier transform infrared (OP-FTIR) optical spectroscopy and the modified combustion efficiency (MCE) (Selimovic et al., 2018). The MCE trace is colored by the key and scale on the right.

Second, we added a Figure S6 to the supplemental material. This figure shows a linear best-fit of the High-temperature and low-temperature time series to the time series of $CO_2$, $NO_x$, and CO. Flaming-combustion compounds CO2 and NOx clearly have a weak relationship to the high- and low-temperature factors. In section 3.1 (line 285), we have added

"**On the contrary, emissions of compounds mainly from flaming or non-pyrolysis smoldering processes, such as CO, CO₂, and NO_x (Figure 1c), do not correlate well with the individual PMF factors (more detailed discussion is given in Section 3.5). This indicates that the two PMF factors do not correspond to the**

flaming and smoldering combustion processes that are described by MCE and often referenced in biomass burning literature. The main source of VOC emissions is pyrolysis of fuel biopolymers, and *not* the flaming and/or other combustion processes. Therefore, we primarily attribute these two factors to high-temperature pyrolysis and low-temperature pyrolysis, respectively, and will use these names to describe these factors in this work."

[Figure]

Figure S6. Linear fits of (a) $CO_2$, (b) $NO_x$, and (c) CO emissions (in ppmv) by the high- and low-temperature pyrolysis time series (in ncps) for Fire #37 (Ponderosa pine realistic mixture). Each plot shows the stacked contributions of the high- and low-temperature factors (shaded area), compared to the measured mixing ratios (solid line). The best-fit equation and correlation coefficient are also provided.

Third, Section 3.5 has been edited and added a new figure (Figure 11). This new figure includes a time-series of MCE and high/low-temperature factors for an example fire; a comparison of instantaneous MCE to high/low-temperature factors across many fires; and a comparison of fire-integrated MCE to high/low-temperature factors across all fires.

The edit reads, "However, MCE does not parameterize the relative amounts of high- and low-temperature pyrolysis products very well, **either instantaneously or on a fire-integrated basis (Figure 11)**." (line 545)

MCE data in Figure 10 has been removed, and now this figure has only the ratio of fire-integrated VOC molar emissions.

[Figure]

Figure 11. The comparison of contribution of high-temperature factor versus modified combustion efficiency (MCE). (a) Time series of Fire #37 (Ponderosa pine realistic mixture). (b) Scatter plot of instantaneous high-temperature contribution versus MCE for all Ponderosa pine fires. (c) Scatter plot of fire-integrated high-temperature contribution versus MCE for all fires. Contribution of high-temperature factor was calculated by $\Sigma VOC_{high-T}/(\Sigma VOC_{high-T} + \Sigma VOC_{low-T})$ instantaneously or on a fire-integrated basis.

**(2)** Next, we address whether the high- / low-temperature description accounts for variation in burn conditions, and how specific burn conditions may quantitatively affect the ratio between the two profiles. One of the strengths of PMF analysis is that we do not need to know all the details of all the processes going on in a fire: PMF groups together all the processes that result in similar VOC emission profiles. A wide range of MCE occurred during the Firelab experiments (newly added Figure 11, Table 1). Because the two PMF profiles describe 85% of variability across all fires, then the two PMF profiles captured the role of various influences as described below.

We did not attempt to develop a quantitative parameterization of the ratio of high-temperature factor to low-temperature factor based on specific fuel conditions (moisture content, surface area, and so on). Such a parameterization is well beyond the scope of this manuscript, but could be a useful target for future work. The fires studied in this experiment incorporated a wide range of fuel moisture content and MCE, so we may reasonably suggest that the high- and low-temperature description of emissions may be applicable under many conditions. We have shown this first by including the fuel moisture content and MCE information in Table 1, which is now in the main text, and by stating the range of moisture content and MCE in section 2.2 (line 168):

"**Fuel moisture content ranged from 0.6% to 55.6%, and instantaneous MCE ranged from 0.75 to 1.**"

We edited section 3.2 (line 388) to read,

"**These many diverse chemical processes are likely happening simultaneously during a fire, and their relative intensities may change based on fuel composition, fuel moisture content, or other as-yet poorly defined parameters. However, the net result of all these variables is the emission of just two major compositional groups.** The VOCs that comprise these two groups mostly consist of the pyrolysis products described above and their analogs. During most of these fires, the emissions of any particular VOC can be described by a linear combination of the high-temperature and low-temperature pyrolysis time series."

We edited the last paragraph in section 3.5 (line 600) to read,

"**The current study incorporated a wide range of MCEs and fuel moisture contents (Table 1), so the two-factor description may be applicable under many conditions. However, some other factors should be required for specific burns, as discussed below.**"

**(3)** Finally, we discuss the relationship between the two PMF-derived profiles and physical processes known to occur in fires. The high-/low-temperature description is analogous to MCE in several ways. MCE describes the overall degree of carbon oxidation; similarly, the high-/low-temperature profiles describe the overall pyrolysis temperature. MCE is experimentally measured by the ratio of CO and $CO_2$, and the high- and low-temperature profiles are measured by the VOC composition. The CO and $CO_2$ ratio is related to the extent of carbon oxidation because less efficient flame-processing results in a relative enhancement of reduced compounds. Similarly, the VOC composition is related to the pyrolysis temperature because higher temperatures enable bond-breaking in biopolymers, leading to smaller molecules at higher temperatures and larger, more substituted molecules at lower temperatures. The relationship between temperature and VOC composition is well established in pyrolysis literature (e.g. Collard and Blin, 2014, Liu et al., 2016). We have edited several sections of the text to clarify.

We have edited section 3.2 (line 339) to read,

"T**he compositional differences between the two profiles can be qualitatively explained by the temperature of the pyrolysis reactions thought to be the main production mechanism of the VOCs, such as depolymerization, fragmentation, and aromatization (Yokelson et al., 1996; Yokelson et al., 1997; Collard and Blin, 2014; Liu et al., 2016). This is illustrated by the relative contributions from the high-temperature versus low-temperature factors for most emitted VOCs. VOCs expected from high-temperature processes have a higher emissions contribution from the high-temperature factor, and likewise for low-temperature VOCs and the low-temperature factor**."

In section 3.5 (line 557), we have added,

"**Our results indicate that VOC emissions are even more closely correlated to the biopolymer composition and the surface-to-volume ratios of fuels, than to the MCE**. "

Additional material has been added to Section 3.5 describing the correlation between low- and high-temperature factors and the measured air temperature, and this is detailed in our response to reviewer comment #3.
* * *
2) The justification for the two factor PMF solution is not at all clear. Figure 3 should be modified to show the average VOC emission profiles of both the high temperature and low temperature factor for a PMF solution of 2 factors, 3 factors and 4 factors respectively, in the same panel (as three different colored lines for each m/z). In addition, the same should also be shown for the third and fourth factors. The issue with the correlation plots given in the supplement is that they mask more than they reveal whereas in a VOC emission profile of the type shown in Fig 3, specific information on which compounds migrate from one factor to a new factor can be clearly made out and is more helpful for assessing whether the proposed solution is justified.

According to comments from both reviewers, we have significantly modified Figures S2 and S3 to more clearly show that a two-factor solution is reasonable, and that additional factors do not add useful information.

Figure S2 now shows the mass spectra from the 2-, 3-, and 4- factor solutions. The two factors that account for most of the variability (Factor 1 and Factor 2) do not change substantially as the number of factors increases. The additional factors in the 3- and 4- factor solutions have similar mass spectra to Factor 1 and Factor 2.

[Figure]

Figure S2. Comparison of mass spectra from 2-, 3-, and 4-factor PMF solutions. The two factors that account for most of the variability (Factor 1 and Factor 2) do not change substantially as the number of factors increases. The additional factors in the 3- and 4-factor solutions have similar mass spectra to Factor 1 and Factor 2.

Figure S3 now shows the time series of a representative fire of Ponderosa pine. The measured signal is compared to the PMF solution with two, three, and four factors. For each solution (2, 3, and 4-factors), the time series of the individual factors are shown as subplots. In each subplot, the linear best fit of the high- and low-temperature factors is shown. It can now be clearly seen that there is low residual with the 2-factors solution; additional factors do not substantially change the residual; and that each factor from the 3- and 4-factor solutions can be described by a linear combination of the high- and low- temperature factors.

**A. PMF factor fit to measured signal**

**B. High- and low-temperature factors linear fit to other possible solutions**

2-factor solution
- High-temp. factor
- Low-temp. factor
- Measured signal

3-factor solution
- Factor 1
- Factor 2
- Factor 3
- Measured signal

F1=0.9264(High-T)
*r* = 0.9988

F2=0.0507(High-T) + 0.7063(Low-T)
*r* = 0.9200

F3=0.0224(High-T) + 0.3466(Low-T)
*r* = 0.8432

4-factor solution
- Factor 1
- Factor 2
- Factor 3
- Factor 4
- Measured signal

F1=0.7966(High-T)
*r* = 0.9862

F2=0.0772(High-T) + 0.5565(Low-T)
*r* = 0.8786

F3=0.4280(High-T)
*r* = 0.9862

F4=0.1904(High-T) + 0.1241(Low-T)
*r* = 0.7137

Figure S3. Comparison of time series from 2-, 3-, and 4-factor solutions. The time-series shown is the total instrument signal from a representative fire of Ponderosa Pine (a part of Fire #37). The individual factors are from the PMF analysis of the extended time series (in which all fires of Ponderosa Pine were concatenated). The left-side plots show the stacked contributions of the factors, compared to the measured signal. The small plots on the right show the time series of the individual factors (solid lines). The high- and low-temperature factors were fit to each factor in the 3- and 4-factor solutions. The best-fit was done using the extended time series. These best-fits are shown as the shaded areas in the right-side plots, and the best-fit equation and correlation coefficient are also provided.
* * *
3) The paper makes generic assertions about low and high temperature emissions, but in none of the experiments was the temperature data provided/shown or even mentioned. I presume the authors have such data /or can provide it. While I understand different parts of the fuel may be subject to different stages of combustion (flaming, combustion, distillation) at the same time and the emissions are a net resultant of these, some temperature data from the burn experiments where a large temperature range was observed from start to finish is certainly in order.

Temperature data was taken during the experiment, via the air temperature of the emissions at the level of the sampling inlet (data from FTIR instrument). The temperature of the fire itself is not easy to define or measure. A thermometer placed inside the fire will only measure the local temperature. An external measurement such as an IR camera may not detect interior parts of the fire that are hidden from view. The air temperature measurement was not always related to the fire conditions because it is strongly affected by the amount of fuel burning relative to the amount of dilution air, the background temperature changed considerably between fires and over the course of a fire, and there was frequently a lag between the onset of emissions and air temperature increase. We have now included a supplementary figure (S8) showing some of this data, for Ponderosa pine fires. There is some correlation between air temperature and the VOC factors. For other fires, the relationship is not clear, likely due to the aforementioned issues with the temperature measurement. Therefore we do not rely on this data to quantitatively parameterize the high- and low-temperature factors.

[Figure]

Figure S8. The comparison of contribution of high-temperature factor versus air temperature of the emissions measured at the sampling inlet of the PTR-ToF-MS. (a) Time series of Fire #37 (Ponderosa pine realistic mixture). (b)-(d) Scatter plots of instantaneous high-temperature contribution versus temperature for Fire #37, #59 (Ponderosa pine realistic mixture), and #38 (Ponderosa pine litter).

At line 559, Section 3.5 has been edited to read,
    "**It is also seen that for some fires the air temperature correlates with the high-temperature contribution (e.g., Fires #37 and #59 shown in Figure S8a-c). This suggests that the VOC emissions are certainly related to the temperature within a fire. However, some other burns did not have a good correlation between the temperature and VOC emissions (e.g., Fire #38 shown in Figure S8d), because the temperature**

**measurement had some issues in the present work: (i) background temperature for each burn was different, (ii) some burns have colder temperature at end compared to start, which means that the laboratory was not controlled at constant temperature, and (iii) the increase in air temperature often lagged behind the emissions, especially at the start of a fire**."

In Section 2.2 (line 172), we have added:
"**The present experiments did not have a direct measurement of temperature within the fire, which is not homogeneous and therefore difficult to define. Rather, the air temperature of the emissions was measured by the FTIR instrument, located at the sampling inlet of the PTR-ToF-MS. The hot gases from the fire were mixed with air from the room, cooling the air significantly, but the trends in temperature are related to the initial temperature of the emitted gases.**"
* * *
4) The authors should clarify/mention the practical steps that need to be followed to employ the high and low temperature pyrolysis profiles for modeling VOC emissions from "many types of biomass burning" as asserted in the abstract and conclusion.

At line 719, the Conclusions section was revised as follows:
"**Our framework provides a way to understand VOC emissions variability in other laboratory and field studies of biomass burning. We highlight two areas of useful future work. First, external tracers should be found that will allow the prediction of the relative contribution of individual profiles. This could include specific chemical species, an understanding of how fuel or burn characteristics relate to the relative contribution of the two profiles, or a relationship between some measure of fire temperature and the VOC profiles. Second, the SOA and ozone formation potential of the two profiles should be determined. With this further work, the VOC profiles could be widely useful to model VOC emissions from many types of biomass burning in the western US, with additions to the framework being needed for fires that burn a lot of duff.**"
* * *
Some specific details:
Line 136-138: Citing Selimovic et al. 2017 for important and relevant details is not sufficient in my opinion. For this paper to be considered as an independent paper, at least a summary of the essential info needs to be provided in the main text. For example, how many fires for each fuel type were sampled in the present work? how similar / dissimilar were the emission profiles when the same fuel type was burnt? Was the pyrolysis temperature or temperature profile during the burns measured? What about the variability induced by other burn conditions?

Table S1, which contains a summary of the fuels burned, which fuel components were burned in each fire, the number of fires of each type, the residuals from PMF, and correlation coefficients of the emission profiles, has been added to the main text in section 2.2 "Fuel and biomass burn descriptions" as Table 1. Additionally, we added MCE and fuel moisture content information to this table. The lines 168-177 have been edited to read,
"**Fuel moisture content ranged from 0.6% to 55.6%, and instantaneous MCE ranged from 0.75 to 1. Additional** details on the fires and fuels are given by Selimovic et al. (2018) including: pre- and post-fire weight, weight of fuel components, and elemental composition (C, H, N, S, and Cl by weight). Each fuel type was burned several times. All fires consumed most of the fuel. **The present experiments did not have a direct measurement of temperature within the fire, which is not homogeneous and therefore difficult to define. Rather, the air temperature of the emissions was measured by the FTIR instrument, located at the sampling inlet of the PTR-ToF-MS. The hot gases from the fire were mixed with air from the room, cooling the air significantly, but the trends in temperature are related to the initial temperature of the emitted gases.**"

[revised manuscript text omitted]

[b] "Duff" data is excluded.
* * *
Further discussion points:

The authors need to elaborate and clarify regarding the limitations of their methodology. In particular, is the interpretation of the data resting just on the statistical treatment, correlation with known products emitted at certain stages of combustion and a hypothesized area/volume ratio? If so, how can these findings be extrapolated to scenarios in the real world which differ from the experimental conditions of the present work? How would other known factors which are missing from their interpretation and analyses such as oxygen (air to fuel ratio variability) and/or moisture content complicate or change the predicted emission behavior? In particular, if emissions resulting from burns of duff and rotten wood are not explained well by their present predictive algorithm, could variability of conditions significantly alter VOC emission profiles of the other currently "well explained" fuels.

To answer this comment, we have added additional information and discussion comparing the range of fire conditions in this study and those reported in the literature, and have elaborated necessary future work and the limitations of this study. The specific edits are as follows.

The third paragraph of Section 3.5 has been edited with additional information (MCE) for the previous field and laboratory studies, in order to show that the field (and previous laboratory) data have large variability in burn conditions:

"Studies of laboratory burns and wildfires have reported variable emission ratios (or factors) for various VOCs **as well as fire-integrated MCE**, even for similar fuel types. Here we investigate how well total VOC emissions in biomass burning can be fit by the average VOC emission profiles (Figure 3) using emission factors and ratios reported in the literature for laboratory and field burns (Gilman et al., 2015; Stockwell et al., 2015; Akagi et al., 2011). When fitting the present high- and low-temperature factors to the other biomass burning data, total VOC emissions can be described with different relative fractions of the factors (Figure S10). For example, the best fit to a laboratory study by Gilman et al. (2015), using fuels from southwestern, southeastern, and northern U.S. (e.g., pine, spruce, fir, chaparral, mesquite, and oak) **with MCE = 0.75-0.98**, includes 32% high-temperature and 68% low-temperature VOC emissions; for another laboratory study by Stockwell et al. (2015) including several types of grass, spruce, and chaparral **with MCE = 0.68-0.99**, 59% high temperature and 41% low temperature; temperate forest fires **(MCE = 0.95)** reported by Akagi et al. (2011), 77% high temperature and 23% low temperature, while in the case of chaparral fires **(MCE = 0.96)**, 48% high temperature and 52% low temperature. The fitting can be done with high correlation coefficient ($r \geq 0.92$) for all the literature data (Figure S10). This is further evidence that at most two factors can explain the majority of VOC variability. Therefore, these two factors could be used to fill in VOCs not measured in the other studies which sometimes had less chemical detail. **The current study incorporated a wide range of MCE and fuel moisture content**

**(Table 1), so the two-factor description may be applicable under many conditions. However, some other factors should be required for specific burns, as discussed below.**"

The Conclusions section was revised to elaborate on future work as follows (noted previously in response to comment #4):

**"Our framework provides a way to understand VOC emissions variability in other laboratory and field studies of biomass burning. We highlight two areas of useful future work. First, external tracers should be found that will allow the prediction of the relative contribution of individual profiles. This could include specific chemical species, an understanding of how fuel or burn characteristics relate to the relative contribution of the two profiles, or a relationship between some measure of fire temperature and the VOC profiles. Second, the SOA and ozone formation potential of the two profiles should be determined. With this further work, the VOC profiles could be widely useful to model VOC emissions from many types of biomass burning in the western US, with additions to the framework being needed for fires that burn a lot of duff."**

The present VOC emission profiles are able to describe the emissions from rotten wood burns; however, an appropriate VOC tracer should be found. For duff burns, the content of nitrogen-containing compounds is higher than for the other fuel components (e.g., canopy and litter), and the VOC composition may depend on the local environment. So, the VOC emissions from duff burns cannot be accurately predicted using the high- and low-temperature VOC profiles. In order to describe that duff burns are limitation of this methodology, a phrase has been added in Conclusions (line 725):

"With this further work, the VOC profiles could be widely useful to model VOC emissions from many types of biomass burning in the western US, **with additions to the framework being needed for fires that burn a lot of duff**."
* * *
MINOR/TECHNICAL COMMENTS
The language and presentation is some parts of the MS can be improved with more careful reading. Two examples are mentioned below: Abstract: Line 26: Precursors to the formation. . .. " the formation of.." is redundant.. Line 27: Measurements collected. . .. Measurements performed?

According to these comments, the corresponding points were fixed.

---

## Author Response (AR2)

**To: Prof. Jacqui Hamilton:**

June 19, 2018

Re: ACP-2018-52

Dear Prof. Jacqui Hamilton:

Thank you for your comments to our paper entitled "High- and low-temperature pyrolysis profiles describe volatile organic compound emissions from western US wildfire fuels". Here the manuscript has been revised according your comments.

**Comment: I am happy that you have dealt with the reviewers comments and will proceed to publication. I have one small change to make prior to acceptance. In the new figure S9- can you add some text to the legend to describe A and B. While I can see it on the top panel, it took me a little while to find it and I think it should be included in the text legend.**

**Answer:** We added a text to describe coefficients $A$ and $B$ in the caption of Figure S9, as follows:

"Figure S9. The comparison of contribution of high-temperature factor versus ethyne/furan ratio. (a) Time series of Fire #37 (Ponderosa pine realistic mixture). (b) Scatter plot of instantaneous high-temperature contribution versus ethyne/furan ratio for all Ponderosa pine fires. (c) Scatter plot of fire-integrated high-temperature contribution versus ethyne/furan ratio for all fires. Contribution of high-temperature factor was calculated by $\Sigma VOC_{high\text{-}T}/(\Sigma VOC_{high\text{-}T} + \Sigma VOC_{low\text{-}T})$ instantaneously or on a fire-integrated basis. **Ethyne/furan ratio was calculated by** $\dfrac{Ethyne/A}{Ethyne/A + Furan/B}$ **instantaneously or on a fire-integrated basis. Coefficients $A$ and $B$ correspond to 0.0393 (in ppbv/total VOC ppbv) for ethyne in the high-temperature factor and 0.0159 (in ppbv/total VOC ppbv) for furan in the low-temperature factor, respectively.**"